# Beta band oscillations in motor cortex reflect neural population signals that delay movement onset

**Preeya Khanna[1], Jose M Carmena[1,2,3]\***

[1]UC Berkeley-UCSF Joint Graduate Program in Bioengineering, University of California, Berkeley, Berkeley, United States; [2]Department of Electrical Engineering and Computer Sciences, University of California, Berkeley, Berkeley, United States; [3]Helen Wills Neuroscience Institute, University of California, Berkeley, Berkeley, United States

**Abstract** Motor cortical beta oscillations have been reported for decades, yet their behavioral correlates remain unresolved. Some studies link beta oscillations to changes in underlying neural activity, but the specific behavioral manifestations of these reported changes remain elusive. To investigate how changes in population neural activity, beta oscillations, and behavior are linked, we recorded multi-scale neural activity from motor cortex while three macaques performed a novel neurofeedback task. Subjects volitionally brought their beta oscillatory power to an instructed state and subsequently executed an arm reach. Reaches preceded by a reduction in beta power exhibited significantly faster movement onset times than reaches preceded by an increase in beta power. Further, population neural activity was found to shift farther from a movement onset state during beta oscillations that were neurofeedback-induced or naturally occurring during reaching tasks. This finding establishes a population neural basis for slowed movement onset following periods of beta oscillatory activity.

**\*For correspondence:** jcarmena@
berkeley.edu

**Competing interests:** The authors declare that no competing interests exist.

## Introduction

Beta band oscillations (typically defined as 13–30 Hz) have been noted to reliably emerge during specific motor actions such as isometric contraction and precision reaching, after movement-related cues, and prior to instructed reaches (*Baker et al., 2003*, *Baker et al., 1997*; *Canolty et al., 2012*; *Engel and Fries, 2010*; *Leventhal et al., 2012*; *Saleh et al., 2010*; *Sanes and Donoghue, 1993*). These findings have prompted hypotheses about what underlying state beta oscillations may be signaling. The most prominent of these hypotheses suggest beta oscillations indicate ongoing sensorimotor integration (*Baker, 2007*), coordination (*Rubino et al., 2006*), idling (*Engel and Fries, 2010*; *Gilbertson et al., 2005*; *Pfurtscheller et al., 1996*), motor preparation (*Donoghue et al., 1998*), or attention (*Fetz, 2013*; *Saleh et al., 2010*). These distinct hypotheses are generated by correlating beta power with phases of different motor tasks. Explaining when beta oscillations emerge and how they reflect underlying neural population computations is challenging given their presence during a multitude of behaviors.

Electrical stimulation is one approach used to experimentally induce oscillations. Slowly oscillating cortical macrostimulation (up to 1.7 Hz) has been shown to entrain single unit neural activity in anesthetized rodents, although this entrainment was overpowered by endogenous rhythms in an awake situation (*Ozen et al., 2010*; *Venkatraman and Carmena, 2009*). Non-invasive stimulation via transcranial alternating current stimulation (tACS) and repeated transcranial magnetic stimulation (rTMS) has been used to induce changes in cortical oscillations (*Fröhlich, 2014*), but the frequency of the

**eLife digest** Behaviors from reading to reaching for objects require large numbers of neurons to fire at the same time. Electrodes on the surface of the scalp, or within the brain itself, can detect this coordinated activity in the form of brain waves, or oscillations. Different regions of the brain produce oscillations with different frequencies. Areas that control movement, known collectively as the motor system, generate oscillations with a frequency of 25 to 40 cycles per second, called beta oscillations. These emerge when individuals are paying attention, processing cues to decide where to move, or waiting to move.

Patients with Parkinson's disease – who suffer from tremors, slowed movements, and have trouble initiating movements – show increased beta oscillations when their symptoms are worst compared to when their symptoms are well controlled. Understanding how beta oscillations affect behavior might therefore reveal how exaggerated beta oscillations contribute to symptoms of Parkinson's disease. However, existing tools for activating or silencing neurons do not allow us to manipulate large groups of neurons with fine enough control to study oscillations precisely.

Khanna and Carmena have now solved this problem by training rhesus macaque monkeys to mentally control their own brain waves. The monkeys, which had electrodes implanted in their brains, learned to increase or decrease their beta oscillations on demand. After achieving the correct level of beta oscillations, the monkeys had to reach towards a target with their arm in order to obtain a reward. Khanna and Carmena found that the monkeys took longer to initiate arm reaches when they had higher levels of beta oscillations. Further, when the monkeys increased their beta oscillations, populations of neurons exhibited patterns of activity that looked very different to the patterns occurring at the start of movement. This suggests that when neurons generate beta oscillations, the motor system is pausing at the expense of being ready to quickly start moving.

This finding meshes well with previous evidence since it is sensible to pause the motor system when paying close attention to a task, deciding where to move, and while waiting. Too much pausing may slow down movements and it may become difficult to initiate them, potentially explaining the link between worsened symptoms and higher beta oscillations seen in Parkinson's disease. Overall, the findings of Khanna and Carmena indicate that beta oscillations reflect brain states in which individuals are pausing, and provide a new tool for studying how oscillations relate to behavior.

induced oscillations was not solely dependent on the stimulation frequency (*Marshall et al., 2006*; *Wach et al., 2013*). Furthermore, the reported changes in motor behavior from stimulation at beta frequencies are inconsistent (*Davis et al., 2012*; *Feurra et al., 2011*; *Pogosyan et al., 2009*).

This study uses neurofeedback to manipulate beta oscillations in subjects' motor cortices with the aim of studying effects on arm reaching. While neurofeedback was pioneered by rewarding changes in firing rates of single motor cortical cells (*Fetz, 1969*), learning to control cortical local field potential (LFP) features has been proposed (*Fetz, 2013*; *Moxon and Foffani, 2015*) and subsequently demonstrated (*Engelhard et al., 2013*; *So et al., 2014*). In this study, three macaque monkeys with chronic microwire electrode arrays implanted in their motor cortices were trained in a sequential neurofeedback and arm-reaching task. Sequential neurofeedback-behavior task designs have been used previously with a variety of neural recording modalities, neural signal features, and behaviors (*Gevensleben et al., 2009*; *McFarland et al., 2015*; *Schafer and Moore, 2011*; *Subramanian et al., 2011*). In this study, a neurofeedback-reaching (NR) task was used in concert with simultaneous, multi-scale, high-count neural recordings to first study how the presence of beta oscillations influences arm-reaching behavior, and second, how underlying neuronal population patterns shift when beta oscillations are generated. It has been suggested that the presence of beta oscillations either locally or distally could influence neuronal computation (*Fries, 2015*; *Reimer and Hatsopoulos, 2010*), as slowly oscillating ephaptic fields have been shown to entrain spiking behavior *in vitro* (*Fröhlich and McCormick, 2010*). However, little evidence exists showing that beta oscillations in the local field potential influence spiking activity through ephaptic mechanisms. Thus, in this study we interpret beta oscillations as a statistic of synchronization of the underlying neural signals, not as

a signal that can independently and causally influence neural spiking via ephaptic effects. We aim to investigate how the underlying neural signals change their encoding during epochs when beta oscillations are observed and do not make claims about the causality of beta oscillations on spiking activity.

There are many proposed behavioral correlates of beta oscillations, but to link oscillations to a behavior rigorously it is necessary to understand how oscillations reflect the underlying neural activity that ultimately drives the behavior. Prior studies investigating neural activity changes during beta oscillations used acute, single-electrode recording preparations and showed that single cells are synchronized to ongoing oscillations but that the strength of this synchronization is unrelated to the involvement of the neuron during the motor task (*Murthy and Fetz, 1996a*). These studies also show individual cells do not change their mean spike firing rate but do exhibit a reduction in spiking variability during oscillations compared to the variability exhibited before the oscillation (*Murthy and Fetz, 1996b*). How might these changes in individual units relate to attention, motor preparation, or idling? Modeling groups have aimed to bridge this gap by showing how beta oscillations could be a signal generated by cells conveying top-down information (*Fries, 2015*; *Lee et al., 2013*), could reflect a pattern of firing activating specific cell assemblies (*Canolty et al., 2012*; *Kopell et al., 2011*), or could reflect specific spatiotemporal recruitment of cells (*Best et al., 2017*; *Rubino et al., 2006*). However, experimental evidence of spiking patterns that accomplish the proposed functions while generating beta oscillationsis lacking. In contrast, if one were to omit the role that beta oscillations may play in motor behavior, there is substantial work linking spiking patterns to specific aspects of motor behavior such as movement onset (*Kaufman et al., 2014*), reaction time (*Afshar et al., 2011*; *Churchland and Shenoy, 2007*), movement angle (*Georgopoulos et al., 1982*; *Lillicrap and Scott, 2013*), and movement speed (*Churchland et al., 2006*; *Moran and Schwartz, 1999*), to list just a few.

Here, we report both changes in motor behavior following performance of neurofeedback during a sequential neurofeedback-reaching task, and a neural population shift that mirrors the observed change in motor behavior. Notably, this shift in neural population was also seen in naturally occurring beta oscillations during reaching tasks, suggesting that beta oscillations reflect a common signature of spiking patterns even in different task contexts. This study ties together existing works on behavioral correlates of beta oscillations with hypotheses of how the motor cortex encodes movement onset through the lens of population-level neural activity.

## Results

### Subjects perform a neurofeedback-reaching task above chance level

To explore how beta oscillations reflect changes in spiking activity and movement parameters, we trained three macaque monkeys to perform a typical center-out arm-reaching task (CO task, *Figure 1a*), and a novel sequential beta neurofeedback arm-reaching task (NR task, *Figure 1b*). Prior to training subjects to perform the NR task, beta frequency band limits used in the neurofeedback portion of the NR task were computed from the CO task. A movement onset-aligned trial-averaged spectrogram from the intracortical recordings in contralateral motor and premotor cortex (e.g. Monkey C in *Figure 1d*) showed that the clearest movement-related desynchronization was in the 25–40 Hz band for all monkeys, consistent with early reports of beta oscillations in the macaque motor cortex (*Baker et al., 1997*; *Murthy and Fetz, 1992*). Thus, the beta band limits for the neurofeedback epoch of the NR task were set at 25–40 Hz.

Subjects were then trained to perform the NR task. Trials were initiated by moving the right arm (co-located with a cursor on a screen) such that the cursor fell within a central target. Holding in the center target initiated the neurofeedback epoch where a beta neurofeedback cursor and one of four possible beta neurofeedback targets appeared on the screen (all blue text in *Figure 1b* falls in the neurofeedback epoch). Subjects modulated endogenous motor cortical local field potential signals to move the vertical position of the beta cursor. Specifically, the cursor was controlled by a spectral estimate of beta band power normalized by a spectral estimate of broadband (1–100 Hz) power (see Materials and methods – Calculation of beta neurofeedback cursor). Indeed, modulation of non-beta frequency bands can influence the position of the beta cursor through the normalization factor, a point discussed in detail later in the experimental findings. When the beta cursor fell in the beta

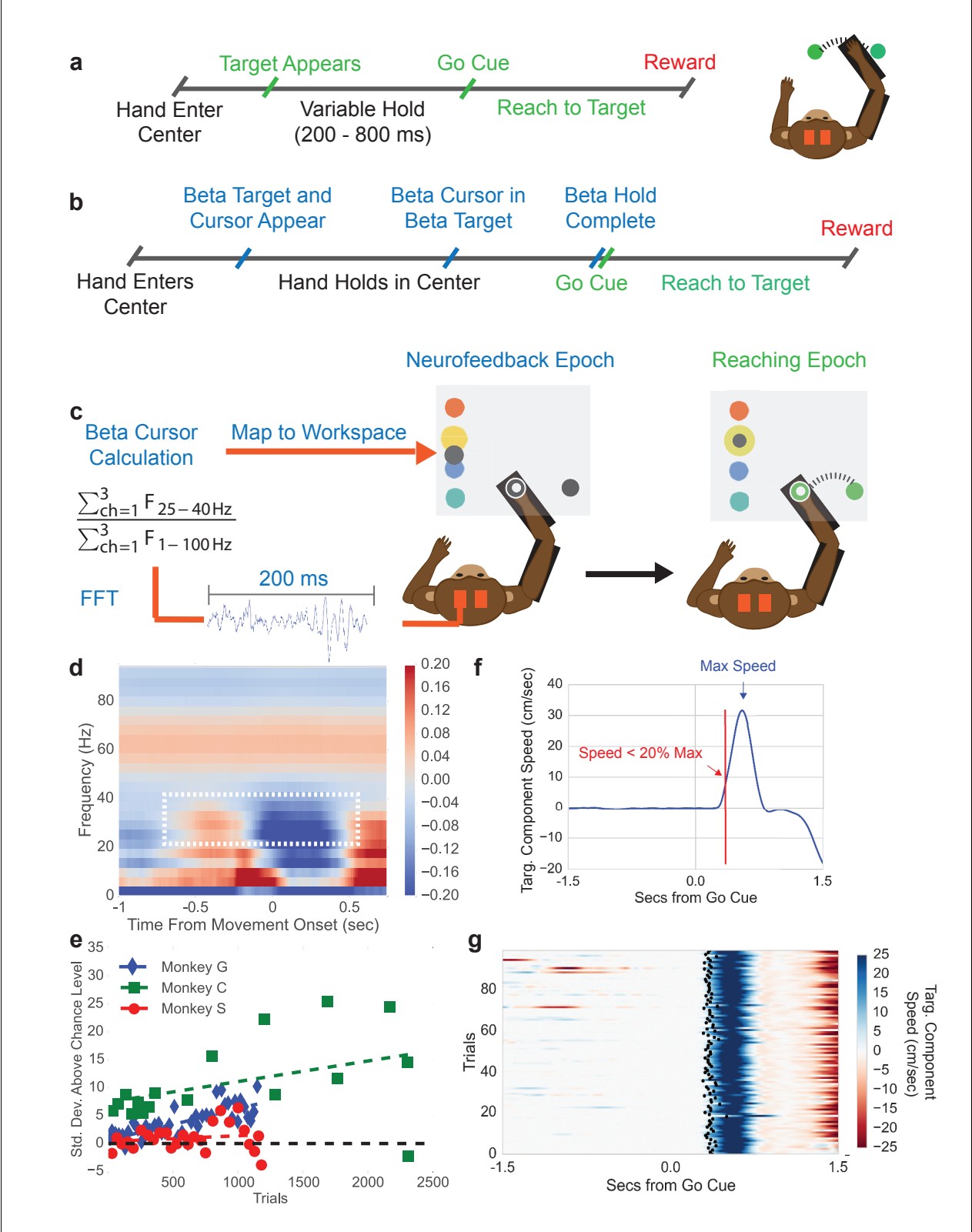

**Figure 1.** Center-out reaching task and neurofeedback-reaching task. (a) Timeline of center-out reaching task (CO task) with variable hold times (200–800 ms) (b) Timeline of neurofeedback-reaching task (NR task), where blue text indicates the neurofeedback epoch and green text indicates the reaching epoch. (c) The NR task feedback loop. Subjects keep their right hand held in a central target throughout the task. They are then shown a single beta target (shown in yellow here) and beta cursor (shown in gray here) on the screen. The cursor has a fixed horizontal position and a vertical

*Figure 1 continued on next page*

*Figure 1 continued*

position that is updated every 100 ms. Once the beta cursor is held in the beta target for 450 ms, the beta cursor and beta target disappear and the subject reaches to a peripheral target 6.5 cm away. (**d**) Trial-averaged spectrogram of movement onset aligned motor cortical LFP signals for Monkey C, with a mean 1/f trend estimated with first-order linear regression and subtracted away. The white box highlights the beta desynchronization in the 25–40 Hz range. (**e**) All three subjects perform the neurofeedback epoch part of the task above chance. The x axis corresponds to all trials from all sessions concatenated. Each point corresponds to a session and each point's position on the x axis corresponds to the first trial that falls within that session. Position on the Y axis indicates standard deviations above mean chance level (shown by the black dotted line). (**f**) Illustration of the metric termed movement-onset time (MOT) throughout the text. Trial-averaged hand speed in the direction of the target is shown in blue with an arrow pointing out the time of maximum hand speed. To find the MOT, step backward in time along the hand speed trace until the hand speed falls below 20% of the maximum speed value. (**g**) 100 trials (rows) of hand speed are shown, where time prior to 0.0 s is the neurofeedback epoch and time after 0.0 s is the reaching epoch in (**b**). Black dots indicate the calculated MOT. Increasing blue corresponds to increasing hand speed.

target, subjects were required to hold within the beta target for 450 ms. After successful completion, both the beta cursor and beta target disappeared, cueing that the reaching epoch had begun (all green text in *Figure 1b* falls in the reaching epoch). Subjects then executed a right-arm reach from the central target to a peripheral target to receive a liquid reward.

When monkeys were first exposed to the task, their time to successfully complete the neurofeedback epoch improved over days. After ~1 week of training per subject, performance plateaued and the beta power to cursor mapping was fixed (see *Table 1*), and subjects completed 3–10 days of executing several 10–40 min sessions of the NR task. Only data from the days with the fixed mapping are presented. During NR task performance from these days, subjects exhibited above chance performance (*Figure 1e*, see Materials and methods - Chance level performance of beta neurofeedback epoch). Monkeys achieved average success rates of 60% and performed on average ~4 successful trials per minute. Errors almost entirely came from accidentally moving the right hand outside the center target during the neurofeedback epoch.

## The neurofeedback epoch results in varying levels of initial beta power prior to reaching

The neurofeedback epoch accomplished the goal of bringing beta power to different levels shown by plotting mean normalized beta power for the last 1 s of the neurofeedback epoch and the first 0.5 s of the reach epoch for rewarded trials to each of the four beta targets (*Figure 2a–c*, averages over all trials, Monkey S: n = 1184, Monkey C: total n = 2328, Monkey G: n = 1042). The first vertical red line indicates the end of the neurofeedback epoch, or go cue for the reaching epoch. The second vertical red line indicates the mean movement-onset time of the reach. The mean normalized beta power of CO trials is shown in black for reference. At the time of the go cue there is a significant group difference in normalized beta power for the four different neurofeedback target conditions (two-tailed Kruskal-Wallis test, Monkey S: n = 1184, H = 47.44, p=2.803e-10, Monkey C: n = 2328, H = 250.1, p<5e-20, Monkey G: n = 1042, H = 48.11, p=2.023e-10).

To assess how subjects co-modulate other frequency bands in addition to the beta band, and to ensure that the beta cursor changes were not a product of the normalization in *Figure 2a–c*, non-normalized, z-scored power spectral densities (PSDs) were computed over the last 0.8 s of the neurofeedback epoch to the first 0.2 s of the reaching epoch (total window is 1.0 s), as shown in *Figure 2d–f*. Mean traces show that high and low beta targets were achieved by increasing and decreasing beta power, respectively. As calculation of the beta cursor position involved an estimate

**Table 1.** Normalized Beta Values Needed to Reach Center of Each Beta Target

| Target location | Monkey S | Monkey C | Monkey G |
| --- | --- | --- | --- |
| High | 0.32 | 0.34 | 0.23 |
| Med-High | 0.25 | 0.26 | 0.18 |
| Med-Low | 0.17 | 0.18 | 0.13 |
| Low | 0.10 | 0.11 | 0.07 |

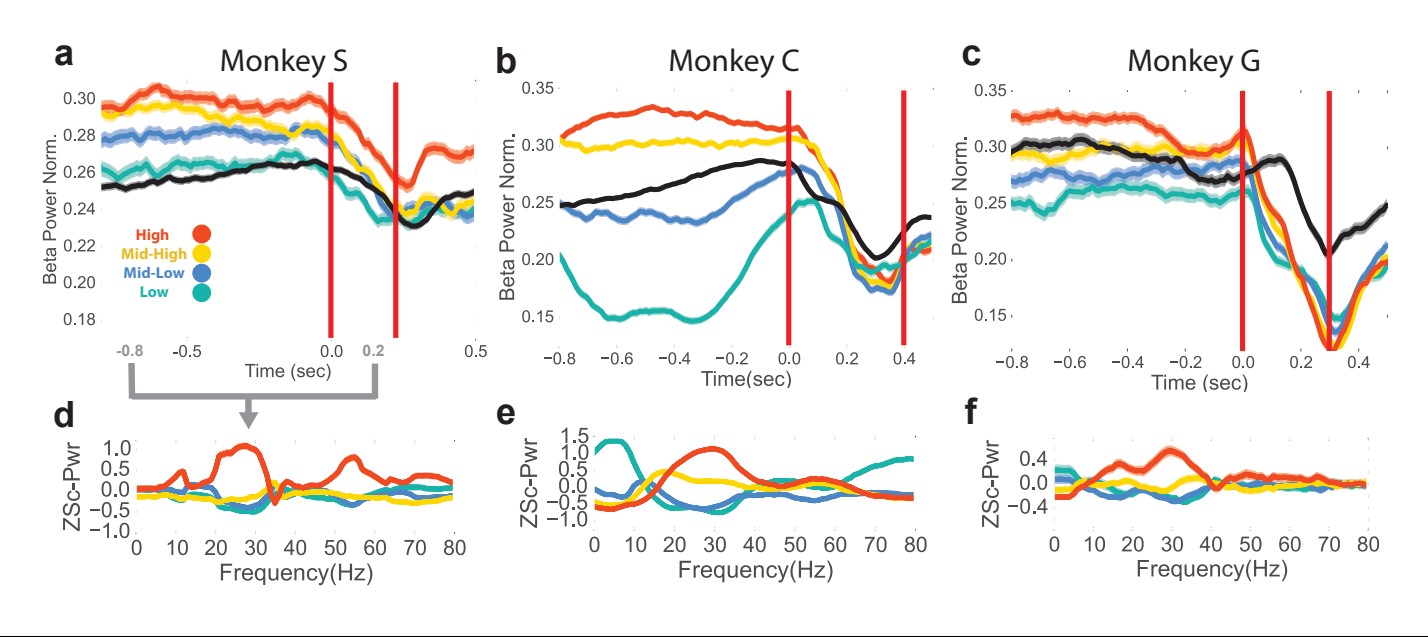

**Figure 2.** Differences in beta power at go cue of reaching epoch in NR task. (**a–c**) Mean (s.e.m) of normalized beta power for Monkeys S, C, and G aligned to the end of the neurofeedback epoch. High, mid-high, mid-low, and low beta targets are in red, yellow, blue, and green, and go-cue aligned CO trials are in black for reference. (**d–f**) Z-scored PSDs estimated from a time slice 0.8 s prior to the end of the neurofeedback epoch (labeled as 0.0 s in a-c) and 0.2 s after the end of the neurofeedback epoch. This time slice is marked in gray below the time axis in (**a**).

of broadband power, changes in non-beta frequencies also affected beta cursor position. In some subjects (Monkeys C, G), increases and decreases in beta power were accompanied by reliable decreases and increases in low frequencies (1–10 Hz).

A final time-domain metric was computed to confirm that the occurrence of beta oscillations differed between the four beta targets in the neurofeedback epoch. Instead of using PSD estimators over a window (as in *Figure 2a–f*), a time-domain method was used to extract time segments with bursts of beta oscillations. Briefly, instantaneous beta amplitude was measured by bandpass filtering the raw LFP with a fifth order Butterworth filter to isolate 25–40 Hz components, and taking the Hilbert transform. If the amplitude exceeded the 60th percentile of beta amplitude (computed each day) for a period of at least 125 ms (3–5 cycles of beta oscillations), the time points within that period were labeled as 'on-beta'. The percentage of time points labeled as on-beta was computed in the same time window as *Figure 2d–f* for all trials. All subjects exhibit increasing percentages of on-beta time points for the low to high beta targets. The mean (s.e.m) of percent of time points labeled as on-beta for low, mid-low, mid-high, high beta targets, respectively, in Monkey S is 9.78 (0.783), 11.2 (0.778), 17.6 (0.913), 39.4 (1.35), in Monkey C is 8.65 (0.523), 17.6 (0.708), 35.9 (0.840), 37.8 (0.868), and in Monkey G is 18.3 (1.04), 16.6 (0.994), 22.5 (1.22), 33.4 (1.43).

These three metrics (normalized beta power in *Figure 2a–c*, non-normalized PSDs in *Figure 2d–f*, and percentage of time labeled as on-beta) demonstrate that the neurofeedback epoch served to increase and decrease beta oscillatory power prior to the arm-reaching epoch.

## Reduced beta power precedes faster movement onset times

In the NR task, movement onset times, movement onset speed, peak reach speed, and movement onset acceleration were calculated for the reaching epoch of each successfully completed trial from days with a fixed beta-to-cursor transform. The two-tailed nonparametric Wilcoxon-like Cuzick's test was used to test for increasing or decreasing ordering of each metric across the four beta target groups (see Materials and methods – Ordering statistics). We tested whether rewarded trials preceded by low, mid-low, mid-high, and high beta targets exhibit increasing or decreasing behavioral metrics. In all three animals, trials with high beta power targets had subsequent reaches with slower

movement onset times (*Figure 3a–c*, two-tailed Cuzick's test, Monkey S: z = 5.763, p=8.267e-09, n = 1183, Monkey C: z = 11.987, p<5e-20, n = 2168, Monkey G: z = 5.856, p=4.729e-09, n = 1028). Note that trials with movement onset times greater than 0.7 s or less than 0.0 s were removed from this and all subsequent analyses (Monkey S: 1 trial, Monkey C: 160 trials, Monkey G: 14 trials). Other groups have found increased beta power correlated with reduced movement onset speed, peak speed, and movement onset acceleration (see *Joundi et al., 2012*; *Pogosyan et al., 2009*). We did not find this consistently across subjects when comparing metrics grouped based on our proxy for beta power, the preceding beta target (see *Table 2*).

## NR task controls

### Beta target acquisition difficulty does not correlate with movement onset time

To ensure the cognitive effort required to increase and decrease beta power during the neurofeedback epoch did not result in increasing movement onset time observed in *Figure 3*, we compared the amount of time it took to acquire each beta target as an approximate measure of each target's difficulty. For Monkeys S and G the time to acquire beta target did not significantly predict MOT in a linear regression, but it did for Monkey C and when data were combined across monkeys (*Figure 4a–c* two-sided Student's t-test for non-zero slope in linear regression, Monkey S: t = 0.7119, p=0.476, n = 1183, Monkey C: t = 2.352, p=0.0188, n = 2168, Monkey G: t = 1.651, p=0.0991, n = 1028, Combined across monkeys: t = 2.0832, p=0.0373, n = 4379). When linear regression was used to predict MOT ($MOT_{pred}$) from time to beta target, and was subtracted from the actual MOT ($MOT_{res}$ = MOT - $MOT_{pred}$), increasing $MOT_{res}$ with increasing beta power target remained (two-tailed Cuzick's test on $MOT_{res}$, Monkey C: z = 13.191, p<5 e-20, n = 2168, Combined data across monkeys: z = 13.615, p<5e-20, n = 4379). Thus, the slight predictive power of time to beta target on MOT does not account for increasing MOT with increasing beta target, as seen in *Figure 3*.

### Screen location of beta targets and reach target location does not account for increasing movement onset time

We examined whether looking at the top part of the screen (where the high power beta target is displayed) required more effort from the subjects with the possibility of slower movement onset times. In Monkey C the relationship between beta target and screen location was reversed by mapping increased beta power to the bottom of the Y-axis for a set of trials analyzed separately. For these trials, the high beta power target became the green Target 1 instead of the red Target 4. *Figure 4e* shows that this manipulation reverses the relationship between target number and MOT (z = −5.971, p=2.354e-09, n = 2113) demonstrating that increasing beta power, not the target position on the screen, consistently correlates with the observed increasing movement onset time.

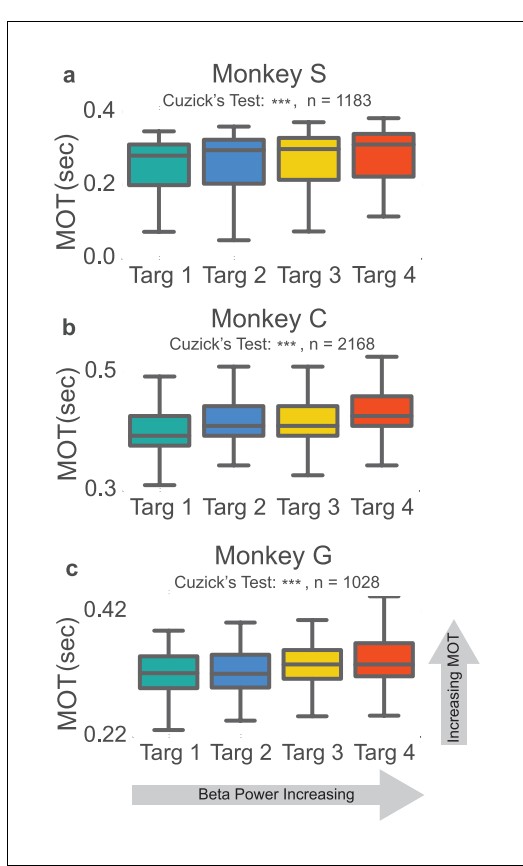

**Figure 3.** Increases in beta power precede increases in movement onset time. (**a–c**) Boxplot of reaching movement onset times grouped by preceding beta target. Subjects exhibit an increase in movement onset time (MOT) when modulating normalized beta power to higher targets. Gray line at center of boxplot is median. ***p<5e-09, Cuzick's two-tailed test.

**Table 2.** Cuzick's Test for ordered trend for reaching kinematics by beta target.

| Metric | Monkey S Z-Statistic (p-value) | Monkey C Z-Statistic (p-value) | Monkey G Z-Statistic (p-value) | Combined Z-Statistic (p-value) |
|---|---|---|---|---|
| Onset Speed | 1.378 (p=0.168) | -3.593 (***) | 0.0734 (p=0.941) | -1.244 (p=0.2134) |
| Peak Speed | 4.476 (***) | 4.142 (***) | -1.330 (p=0.187) | 1.834 (p=0.06670) |
| Onset Acceleration | 1.901 (p=0.0573) | 2.573 (*) | 0.159 (p=0.873) | 0.818 (p=0.4135) |

$^{*}$p<0.05, **p<0.01, ***p<0.001, Cuzick's two-tailed test for ordered grouping. Monkey S: n = 1183, Monkey C: n = 2168, Monkey G: n = 1028, Combined: n = 4379

To test whether the beta target versus movement onset time relationship generalizes to more than a single arm-reaching target, Monkey S performed the original NR task, but with a different arm-reaching target location. Changing the arm-reaching target location produces the same effect (***Figure 4d***, Monkey S: z = 3.972, p=7.117e-05, n = 735).

## Movement onset time increase is specific to beta band frequencies

### Using other methods to compute beta power shows the same movement onset relationship

To confirm that the correlation between lower beta power targets and faster movement onset times did not result from use of a beta cursor calculation method that normalizes beta power by total broadband power, offline we sought to account for the increase in movement onset time using a beta power calculation method that was non-normalized. Non-normalized beta power was computed in the window spanned by the last 0.8 s of the neurofeedback epoch. Trials were then re-labeled according to the quartile in which their non-normalized beta power fell (e.g., if the non-normalized beta power falls in the 0–25th percentile of all trials, the trial would be assigned to group 1). Trials with movement onset times less than 0 s or greater than 0.7 s were removed, as before. The mean movement onset time for each of these new groups is plotted for each monkey (***Figure 5a–c***, darkest and lightest bars correspond to lowest and highest non-normalized power, respectively). All three monkeys exhibit significantly increasing movement onset times with increasing non-normalized beta power (two-tailed Cuzick's test, Monkey S, z = 7.162, p=7.945e-13, n = 1183, Monkey C, z = 7.767, p=7.994e-15, n = 2168, Monkey G, z = 7.709 p=1.266e-14, n = 1028, Combined across monkeys: z = 6.168, p=6.924e-10, n = 4379).

The same trial re-labeling procedure was performed except instead of relabeling by non-normalized beta power, trials were re-labeled by the percentage of on-beta time points in the last 0.8 sec of the neurofeedback epoch using the previously explained time-domain method. Indeed, the same increase in movement onset time follows where trials with a larger percentage of on-beta time points exhibit slower movement onset times (two-tailed Cuzick's test, Monkey S, z = 7.575, p=3.597e-14, n = 1183, Monkey C, z = 5.488, p=4.068e-08, n = 2168, Monkey G, z = 7.890, p=3.108e-15, n = 1028, Combined across monkeys z = 5.1301, p=2.895e-07, n = 4379). Thus, the relationship between increasing beta target and increasing MOT is not dependent on the normalized beta power computation method used in the NR task, and withstands when using a non-normalized or a time domain beta power computation method.

### Non-beta frequencies are co-modulated during the beta neurofeedback epoch of task

As the neurofeedback epoch requires control of normalized beta power, it is possible for subjects to have neurofeedback strategies that involve modulation of non-beta frequency bands to move the cursor. Using the same trial re-labeling procedure as described above, individual trial labels were re-assigned depending on normalized power in non-beta frequency bands (1–10 Hz, 10–25 Hz, 40–65 Hz, and 65–100 Hz) for the same time window as above. The resulting movement onset times are plotted by re-labeled group, frequency band, and monkey in ***Figure 5d–f***. Increasing 10-25 Hz power correlates with increasing MOT (two-tailed Cuzick's test, Monkey S 10–25 Hz: z = −0.6660, p=0.5054, n = 1183, Monkey C 10–25 Hz: z = 5.717, p=1.082e-08, n = 2168, Monkey G 10–25 Hz:

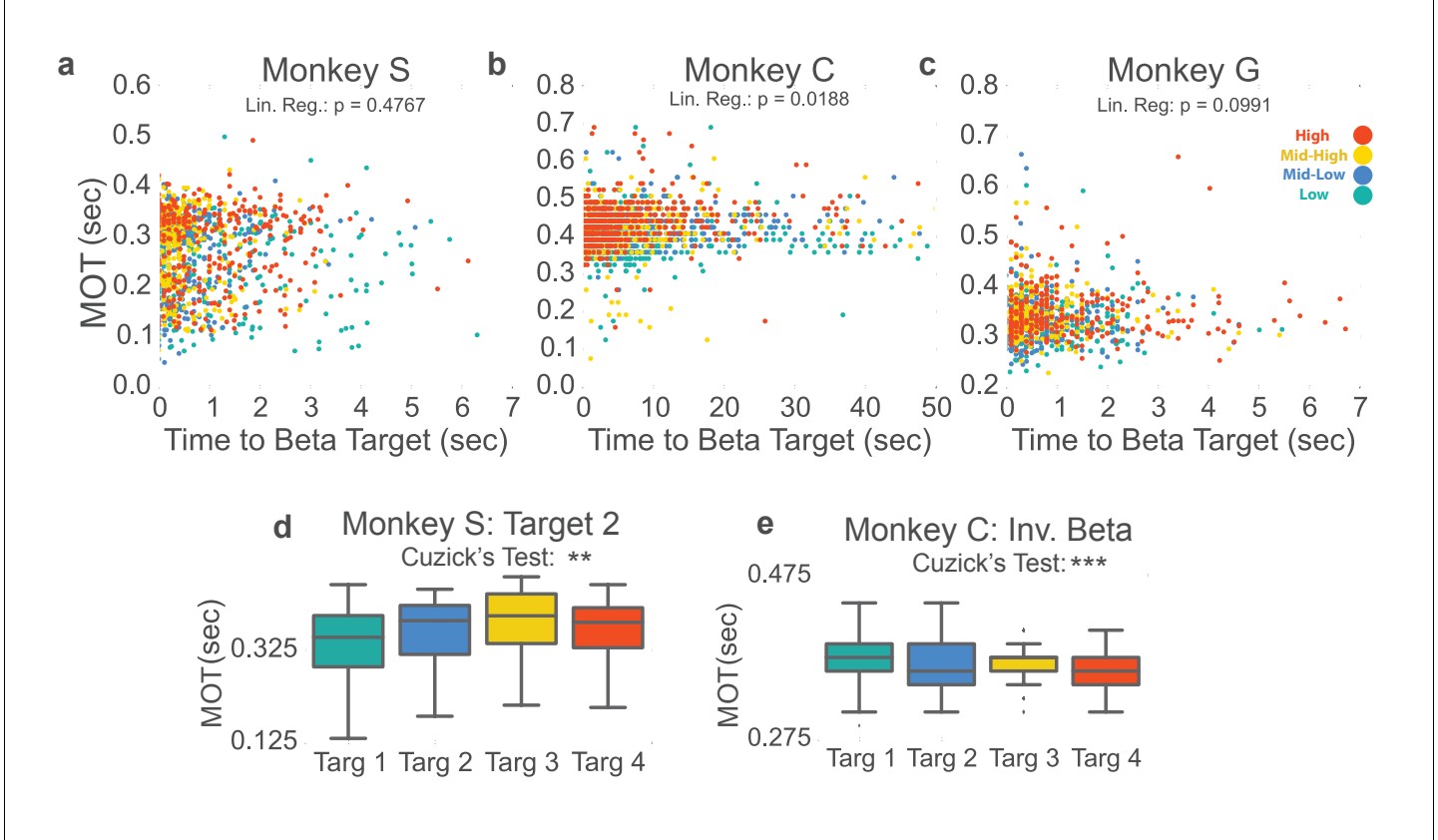

**Figure 4.** Beta target difficulty, beta target location, and arm-reaching location do not account for increase in movement onset times. (**a–c**) Movement onset time (MOT) from *Figure 3* is plotted against time to beta target for Monkeys S, C, and G. Colors correspond to the beta target for that trial following the same colormap as *Figures 2* and *3*. Linear regression is performed to assess whether time to beta target is predictive of MOT. Non-significant p-values for Monkeys S and G show that time to beta target (interpreted as beta target difficulty) does not significantly predict MOT. Monkey C exhibits a significant relationship, but when $MOT_{res}$ is computed by subtracting predicted MOTs from time to beta target from actual MOT, $MOT_{res}$ exhibits the same increase with increasing beta target as seen in *Figure 3*. (**d**) Changing location of the arm-reaching location from 6.5 cm to the right of the central target, to 6.5 cm above the central target does not change the observed relationship between increasing movement onset times and increasing beta power target in Monkey S. (**e**) Inverting the vertical ordering of beta targets on the screen to make green the high-beta target, blue the mid-high beta target, yellow the mid-low beta target, and red the low beta target also shows the same increase in MOT with increased beta target as in *Figure 3b*. *p<0.05, **p<0.01, ***p<0.001, Cuzick's two-tailed test for significant increases and decreases by grouping except where noted.

z = 5.477, p=4.33e-08, n = 1028, Combined across monkeys 10–25 Hz: z = 2.9728, p=2.951e-03, n = 4379). The consistent ordering in the 10–25 Hz band (increased power correlated with increased MOT) is likely caused by the natural beta band for each animal extending into frequencies below 25 Hz. *Figure 2d–f* shows that Monkey C and Monkey G exhibit power trends in frequencies lower than 25 Hz that match those in the 25–40 Hz range. The 65–100 Hz (gamma) band does exhibit consistently decreasing power for higher beta targets across monkeys (two-tailed Cuzick's test, Monkey S: z = −5.0279, p=4.96e-07, n = 1183, Monkey C: z = −4.227, p=2.368e-05, n = 2168, Monkey G: z = −1.775, p=0.0759, n = 1028, Combined across monkeys z = −3.1079, p=1.884e-03, n = 4379). Indeed, beta power and gamma power have been shown to be anti-correlated in motor-related regions during tasks involving movement (*Courtemanche et al., 2003*; *Schoffelen et al., 2005*), in prefrontal cortex during working memory tasks (*Lundqvist et al., 2016*), and in parkinsonian subjects at rest (*Fogelson et al., 2005*). Increased gamma power could be a physiological pattern that emerges with reduced beta power. It is unlikely that subjects are relying on changes in gamma power, which would change the denominator term in the beta cursor computation, to drive their neurofeedback strategy as gamma power constitutes less than 3% of the total broadband estimate (Table 4).

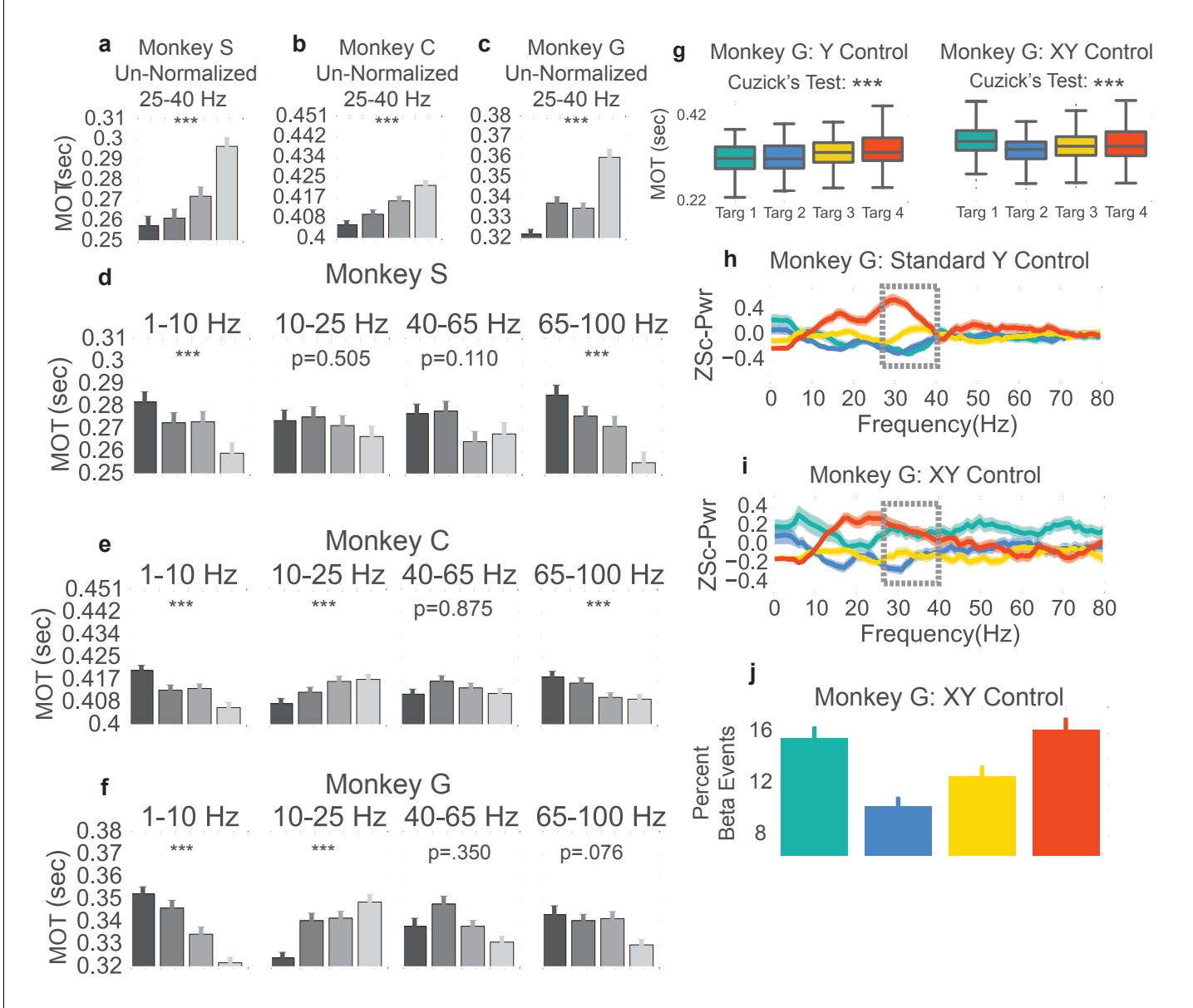

**Figure 5.** Non-beta frequencies' involvement in movement onset time. (**a–c**) Trials from Monkeys S, C, G were re-labeled as low, mid-low, mid-high, and high according to the non-normalized beta power during time slice −0.8 to 0.0 s with respect to the end of the neurofeedback epoch. The movement onset times (MOTs) of the re-labeled trials were compared and the mean (s.e.m) are plotted in each subplot. Below titles, p-values are shown for Cuzick's test. (**d–f**) The same as (a-c) except using normalized non-beta frequencies indicated at the top of plots. (**g**) Right shows MOTs for version of task where the X axis position of the cursor is controlled by normalized 1–10 Hz power in addition to the Y axis position controlled by normalized 25–40 Hz power. Note that Cuzick's test for the right plot assumes ordering of targets is mid-low, mid-high, high, low (instead of low, mid-low, mid-high, high). Left, same MOT plot for Monkey G from *Figure 3c* for comparison. (**h, i**) Z scored PSD plots (same method as *Figure 2d–f*) for different beta targets in the standard beta neurofeedback task (**h**), and the neurofeedback task that incorporates 1-10 Hz power modulation to control X axis position (**i**). Dotted lines indicate that the ordering of targets in the beta range follows the movement onset time ordering in (**g**). (**j**) Percent of time points within the last 0.8 s of the neurofeedback epoch that are part of beta oscillatory episodes during the task requiring both beta and 1–10 Hz control. *p<0.05, **p<0.01, ***p<0.001, Cuzick's two-tailed test for ordered grouping.

The correlation between increased gamma power and reduced MOT was further investigated using model selection analysis. MOTs were linearly estimated using normalized gamma power as a predictor (Model 2, *Table 3*), or normalized beta power as a predictor (Model 1, *Table 3*), or both normalized gamma power and normalized beta power as predictors (Model 4, *Table 3*) from the last

**Table 3.** Predicting MOT with beta ($\beta$, 25-40 Hz), gamma ($\gamma$, 65-100 Hz), and non-beta frequency Low Frequency (LF, 1-10 Hz) bands.

| | $R^2$ Model 1: MOT ~ $\beta$ | $R^2$ Model 2: MOT ~ $\gamma$ | $R^2$ Model 3: MOT ~ LF | $R^2$ Model 4: MOT ~ $\beta+\gamma$ | $R^2$ Model 5: MOT ~ $\beta+$LF | F statistic (p-value): Model 4 > Model 2 | F statistic (p-value): Model 5 > Model 3 |
|---|---|---|---|---|---|---|---|
| Monkey S | 0.0403 | 0.01483 | 0.006417 | 0.05806 | 0.04321 | $F_{(1181, 1180)} = 54.162$, p < 1e-16 | $F_{(1181, 1180)} = 45.373$, p < 1e-16 |
| Monkey C | 0.0138 | 0.004384 | 0.01253 | 0.01733 | 0.01456 | $F_{(2166, 2165)} = 28.531$, p < 1e-16 | $F_{(2166, 2165)} = 4.47718$, p < 1e-16 |
| Monkey G | 0.0767 | 0.005467 | 0.05866 | 0.10561 | 0.07963 | $F_{(1026, 1025)} = 114.77$, p < 1e-16 | $F_{(1026, 1025)} = 23.356$, p < 1e-16 |
| Combined Monkeys | 0.04363 | 0.01383 | 0.018204 | 0.05677 | 0.04522 | $F_{(4377, 4376)} = 199.20$, p < 1e-16 | $F_{(4377, 4376)} = 123.83$, p < 1e-16 |

0.8 s of the neurofeedback epoch. The normalized beta power model explained more MOT variance than the normalized gamma power model (*Table 3*), and the F-test demonstrated that adding normalized beta power as a predictor in a model with normalized gamma power resulted in significant improvement (Model 2 vs. Model 4, *Table 3*). Thus, while gamma power is negatively correlated with MOT, beta power explains more MOT variance than gamma power, and addition of beta power to a model predicting MOT with gamma power significantly improves prediction.

The 1–10 Hz band also shows a consistent across-monkey decrease in movement onset time with increased power, discussed further below (two-tailed Cuzick's test, Monkey S z = −4.290, p=1.785e-05, n = 1183, Monkey C: z = −6.8774, p=6.097 e-12, n = 2168, Monkey G: z = −8.4548, p=2.795e-17, n = 1028, Combined across monkeys z = −5.3049, p=1.127e-07, n = 4379).

## Modified beta neurofeedback task shows that 1–10 Hz band power does not account for movement onset time increase

The 1–10 Hz power subplot of *Figure 5d–f* shows reduced movement onset time with increasing 1–10 Hz power for all three subjects. To investigate whether the movement onset time increase observed truly resulted from changes in beta power and not changes in the 1–10 Hz band power, we performed an experimental manipulation as well as a regression analysis, as above with gamma power. In the experimental manipulation, Monkey G performed a NR task variant where beta power continued to move the beta cursor up and down the Y axis, but now instead of having a fixed X axis position, 1–10 Hz power controlled the cursor on the X axis. The targets were in the same positions as in the standard NR task, but now Monkey G had to ensure that his 1–10 Hz power was neither too low nor too high, or else the horizontal position of the cursor would not fall within the fixed width of the beta target. Monkey G learned this task and after 3–4 days of practice achieved similar performance to the standard NR beta-only task of 5–10 s to each beta target (see Materials and methods - NR task variant using beta band and 1–10 Hz)

In the beta task variant, Monkey G adopted a new strategy for getting to the lowest target. *Figure 5h and i* shows PSDs from the last 0.8 s of the neurofeedback epoch to the first 0.2 s of the reach epoch. For the lowest (green) target in the beta task variant, Monkey G managed to increase the power of his beta frequencies to similar levels as the highest beta target (red) but as he concomitantly increased the power of other frequencies, the denominator term in the normalized beta metric increased more, making the cursor move downwards (*Figure 5i*). To ensure that the PSD plot

**Table 4.** Percentage of broadband power estimate comprised by 65–100 Hz.

| Monkey | Mean (std) Percentage |
|---|---|
| Monkey S | 2.43 +/- 1.13% |
| Monkey C | 1.50 +/-0. 518% |
| Monkey G | 2.14 +/-0. 825% |

reflected the presence of beta oscillations, we also calculated the percent of on-beta time points during the neurofeedback epoch of the NR task variant using the previously explained time-domain method (*Figure 5j*). This metric reflects the same ordering found in the PSD; the lowest beta target had a comparable percentage of on-beta timepoints to the highest beta target.

This task variant effectively decoupled beta and 1–10 Hz power. In the original task, beta and 1–10 Hz power were inversely correlated (low beta power occurred with high 1–10 Hz power and vice versa) but in this modified task, high 1–10 Hz power and high beta power co-occurred during the low, green target and low 1–10 Hz power and high beta power co-occurred during the high, red target. We analyzed whether movement onset times followed the beta power or the 1–10 Hz power ordering. The movement onset times (*Figure 5g*, right) for the green target rose to match the movement onset times of the red target, indicating that the movement onset times followed beta power ordering, not 1–10 Hz power ordering (one-tailed Cuzick's test for significant ordering of beta targets in *Figure 5g*, right assesses increasing movement onset times per the group order 2, 3, 4, 1 instead of group order of 1, 2, 3, 4 used in all other Cuzick's tests. Monkey G: z = 7.1359, p=4.807e-13, n = 1164). If the 1–10 Hz power target ordering is used, then MOTs show no significant trend (one-tailed Cuzick's test for MOTs increase with decreasing 1–10 Hz power, group order of 1, 2, 3, 4, Monkey G: z = −2.5713, p=0.9949, n = 1164). Although *Figure 5d–f* shows a strong co-modulation of 1–10 Hz frequencies with beta frequencies, the 1–10 Hz band does not explain the observed ordering of movement onset times. In addition to the above experimental manipulation, we also assessed the contribution of the 1–10 Hz band in explaining MOT variance. MOTs were linearly estimated using the normalized 1–10 Hz power as a predictor (Model 4, *Table 3*), normalized beta power as a predictor (Model 2, *Table 3*), or both normalized 1–10 Hz power and normalized beta power as predictors (Model 5, *Table 3*). The normalized beta power model explained more MOT variance than the normalized 1–10 Hz power model (*Table 3*), and the F-test demonstrated that adding normalized beta power as a predictor in a model with normalized 1–10 Hz power resulted in significant improvement (Model 4 vs. Model 5, *Table 3*). Thus, while 1–10 Hz power is negatively correlated with MOT, beta power explains more MOT variance than 1–10 Hz power, and addition of beta power to a model predicting MOT with 1–10 Hz power significantly improves prediction.

Finally, we investigated which sub-frequency bands within the 1–10 Hz band were most closely correlated with MOT. We divided the low frequencies into the delta band (1–3 Hz), theta band (4–7 Hz), and alpha band (8–12 Hz). By performing the same analysis as in *Figure 5d–f* with the narrower bands, we found that delta and theta bands, but not the alpha band, strongly correlate with MOT in all three animals (Monkey S: 1–3 Hz: z = −3.276, p=1.052e-03, n = 1183, 4–7 Hz: z = −3.245, p=1.174e-03, n = 1183, 8–12 Hz: z = −1.309, p=0.1907, n = 1183, Monkey C: 1–3 Hz: z = −6.986, p=2.821e-12, n = 2168, 4–7 Hz: z = −6.133, p=8.602e-10, n = 2168, 8–12 Hz: z = −1.779, p=0.0753, n = 2168, Monkey G, 1–3 Hz: z = −8.201, p=2.379e-16, n = 1028, 4–7 Hz: z = −6.166, p=7.024e-10, n = 1028, 8–12 Hz: z = 0.0669, p=0.9467, n = 1028, Combined over monkeys: 1–3 Hz: −5.006, p=5.55e-07, n = 4379, 4–7 Hz: z = −4.355, p=1.33e-05, n = 4379, 8–12 Hz: z = −0.9782, p=0.328, n = 4379).

## Comparing single- and multi-unit responses to beta oscillations during the CO and NR tasks

If the neurofeedback-induced beta oscillations are qualitatively the same as naturally occurring beta oscillations during reaching tasks, it might be unsurprising that increasing beta power biases subjects toward slower movement onset based on previous studies. We investigate exactly how similar the beta oscillations in the different tasks are through the lens of unit neural activity.

On most days, subjects performed 5–10 min of the CO task prior to beginning the NR task. Only days when the CO task was performed were used for subsequent analysis (Monkey G: 6 days, Monkey C: 4 days). Simultaneous single-unit and multi-unit activity were recorded throughout Monkey G's task sessions, and channel level activity (*Chestek et al., 2011*) were recorded throughout Monkey C's task sessions. Both single-unit and multi-unit activity were manually sorted, whereas channel level activity used the auto-thresholding function in Omniplex software (see Materials and methods – Surgery, electrophysiology, and experimental setup). In subsequent analyses, time bins (100 ms or 25 ms depending on analysis) will be labeled as on-beta or off-beta referring to whether they fall within or outside a beta oscillation (same definition used previously–periods in which beta amplitude is above the 60th percentile for at least 125 ms). Bins will also be referred to as slow or fast referring

to whether the mean hand speed within the bin is below or above 3.5 cm/sec. The 'slow' versus 'fast' bin distinction was made to separate bins before movement onset from those after movement onset (approximate movement onset time occurred when hand velocity crossed 3.5 cm/sec). The point in the trial corresponding to the cue for movement onset is referred to as the go cue in both tasks (corresponding to the end of neurofeedback epoch in the NR task).

We assessed whether units fire at similar rates during CO task beta oscillations and NR task beta oscillations. Go cue aligned trials were aggregated for each task, with each trial lasting 2.5 s (1.5 s before go cue through 1.0 s after go cue). Unit activity was binned into 100 ms bins, yielding 25 bins per trial. For every trial, bins that were labeled as slow and on-beta were selected. The distribution of spike counts for these slow, on-beta bins from the NR task was compared with the slow, on-beta bins from the CO task. Counts from fast bins were not used in the analysis because there were very few fast bins that were also on-beta.

Example mean firing rates of four consecutive on-beta time bins in a row, aligned to onset of on-beta, are shown for two example single unit recordings (unit 101a and unit 1a) from Monkey G in *Figure 6a*, where red is the mean firing rate for NR slow, on-beta bins, and blue is the mean firing rate for CO slow, on-beta bins. *Figure 6b* shows the fraction of units exhibiting significantly different mean firing rates between the slow, on-beta bins from the two tasks on each day (Mann–Whitney test, $p<0.05$, number of units recorded per day displayed above bar). Each day, 40–60% of units from Monkey G (15–20% of units from Monkey C, inset) exhibited significantly different firing rates for NR versus CO slow, on-beta bins.

While many individual cells showed changes in mean firing rate during slow, on-beta bins across the two different tasks, it is possible that units could still exhibit a consistent spike rate change in response to beta amplitude changes. We used methods adapted from *Canolty et al. (2012)* to fit a beta amplitude-to-spike rate mapping for the NR and CO tasks to determine whether the units' beta amplitude-to-spike rate correlations are consistent across tasks (see Materials and methods – Beta amplitude-to-spike rate mapping). Briefly, the logarithm of instantaneous beta amplitude was calculated for the entire 2.5 s trial and then correlated against the firing rate of each cell. Three example units are shown in *Figure 6c–e*, where the red and blue traces are the relationship between cell firing and beta amplitude for the NR and the CO tasks, respectively. Some units exhibit similar mean firing rates but different beta amplitude-to-spike rate slopes (*Figure 6c*), some exhibit different mean firing rates but similar beta amplitude-to-spike rate slopes (*Figure 6d*), and some exhibit different mean firing rates and different beta amplitude-to-spike rate slopes (*Figure 6e*).

To assess whether units exhibit similar beta amplitude-to-spike rate slopes across the tasks, we compared within-task and across-task slope estimates. First, two non-overlapping subsets of the CO ($CO_1$, $CO_2$) and NR ($NR_1$, $NR_2$) tasks were used to estimate separate beta amplitude-to-spike rate slopes per unit ($CO_{1, slope}$, $CO_{2, slope}$ and $NR_{1, slope}$, $NR_{2, slope}$). Note that slightly overlapping CO sets were used for Monkey C because of limited CO data (see Materials and methods: Beta amplitude-to-spike rate mapping). Then, the two slope estimates for each task were correlated to assess within-task slope estimate stability (*Figure 6f*: $CO_{1, slope}$ vs. $CO_{2, slope}$ and *Figure 6g*: $NR_{1, slope}$, vs. $NR_{2, slope}$). In *Figure 6f–g*, each plotted marker corresponds to a unit and its color corresponds to the day on which it was recorded (Monkey G: main plot, Monkey C: inset, note same colormap as *Figure 6b*). The printed correlation coefficients are the mean correlation coefficient across days and describe how well linear regression captures the correlation between slope estimates from subset 1 vs. slope estimates from subset 2. For both tasks, correlation coefficients exceed 0.8. These high correlation coefficients suggest a stable within-task beta-to-rate mapping. Across task slope estimates are visually compared in *Figure 6h* ($CO_{all\ data}$ vs $NR_{all\ data}$). In contrast to the stable within-task slope estimates, across task slope estimates are less correlated across units. To assess whether the within-task and across-tasks slope differences are significant, a paired Student's t-test was performed to assess the differences between within-task and across-tasks on $CO_1$ and $CO_2$ slopes, $NF_1$ and $NF_2$ slopes, and $CO_2$ and $NF_1$ slopes, where units from each day were treated as independent observations (Monkey G: $CO_1$ vs. $CO_2$, $t = 1.275$, $p=0.2032$, $n = 355$ units, $NF_1$ vs. $NF_2$, $t = 0.0169$, $p=0.9866$, $n = 355$ units, $CO_2$ vs. $NF_1$, $t = 2.3403$, $p=0.0198$, $n = 355$, Monkey C: $CO_1$ vs. $CO_2$, $t = -1.5075$, $p=0.1323$, $n = 513$ units, $NF_1$ vs. $NF_2$, $t = -1.309$, $p=0.1910$, $n = 513$ units, $CO_2$ vs. $NF_1$, $t = 9.880$, $p=3.473e-21$, $n = 513$, Combined across monkeys: $CO_1$ vs. $CO_2$, $t = 0.6258$, $p=0.5316$, $n = 868$ units, $NF_1$ vs. $NF_2$, $t = -0.5547$, $p=0.5793$, $n = 868$ units, $CO_2$ vs. $NF_1$, $t = 5.1782$,

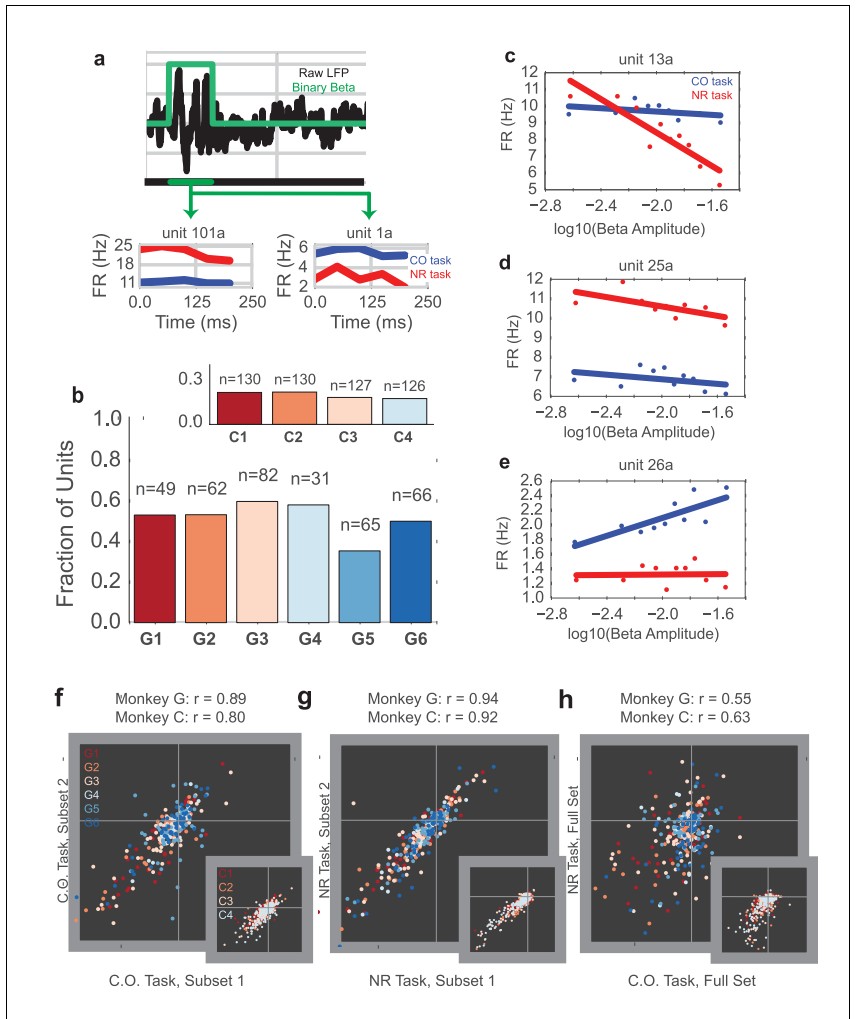

**Figure 6.** Units exhibit different mean firing rates during CO and NR tasks on-beta timepoints and different beta amplitude-to-firing rate mappings during entire CO and NR trials. (a) Schematic of an LFP trace (in black) with an on-beta vs. off-beta indicator shown in green. During oscillatory events, example mean firing rates are shown for two example single units from Monkey G (unit 101a, unit 1a), where the red trace is for on-beta bins during the neurofeedback epoch and the blue trace is for slow, on-beta bins during the CO task. Graphs are aligned to starting bin of beta event. (b) For each day (main plot Monkey G: days G1 – G6, subplot Monkey C: days C1 – C4), a bar plot indicates the fraction of units that exhibit significantly different firing patterns during slow, on-beta time points in the CO and NR task assessed by the Mann–Whitney U test (p<0.05). Number of units recorded per day are printed above each bar. (c–e) Example beta amplitude-to-spike rate mappings for single-units from a day. Mappings in red are from the CO task. Mappings in blue are from the same unit on the same day during the NR task. Units can exhibit similar means but different slopes (c), similar slopes but different means (d), or different slopes and different means (e). (f, g) Stability of beta-to-rate slope estimates from subset #1 versus subset #2 of CO (f) and NR (g) tasks (main plot Monkey G, subplot Monkey C). R values indicate correlation coefficient between beta-to-rate slopes computed from subset #1 and from subset #2, averaged across days. (h) Comparison of beta-to-rate slopes from CO versus NR tasks (full set, not subset). R values indicate correlation coefficient between slopes from CO task versus slopes computed from NR task, averaged across days.

p=2.786e-07, n = 868 units). The subset comparison of $CO_2$ vs. $NF_1$ was randomly chosen for reporting – $CO_2$-$NF_2$, $CO_1$-$NF_1$, and $CO_1$-$NF_2$ showed similar differences in CO vs. NF slopes.

Thus, the mean firing rates of units during slow, on-beta timepoints and the continuous beta amplitude-to-spike rate mappings across tasks are different for many units. Given that the behavioral effect from neurofeedback-induced beta oscillations matches well with hypotheses claiming a

movement-slowing role of beta oscillations during natural movements (*Gilbertson et al., 2005*), it was surprising that individual unit responses during beta activity were so different across tasks.

## Beta oscillations during the NR task and CO task reflect a shift in neural population activity away from a movement onset state

While many individual units exhibit different spiking patterns during beta oscillations in the CO and NR tasks, population-level activity across the two tasks could still exhibit consistent patterns. Specifically, as increased beta oscillations during the NR task are correlated with slower MOT times, beta oscillations may reflect a shift in neural population patterns in a way that affects movement onset. To assess how similar or different CO and NR population patterns are with respect to movement onset, and how the presence of beta oscillations affects these patterns, we trained a classifier on a single day's CO population spiking activity to discriminate bins occurring pre and post movement onset (PreMO, PostMO). We then tested the classifier's performance on the same day's NR population spiking activity to assess first whether the same CO-trained classifier successfully distinguished PreMO and PostMO in the NR task, and second how the presence of beta oscillations influenced the separation of the two labeled classes.

The approach used to discriminate PreMO and PostMO neural population activity was to train a logistic regression classifier on the first two-thirds of the CO spike counts. CO spiking activity was binned in 25 ms, each unit was z-scored according to its mean and standard deviation during the CO task, and each 25 ms bin was labeled PreMO or PostMO. A logistic regression classifier was trained on the binned spike counts with an additional two bins of history (number of spike features per observation equal to three times the number of neurons, and each three-bin observation is referred to as a 'chunk'). The trained classifier yields the probability of each chunk being PreMO and PostMO. By setting a threshold on these probabilities we can assign predicted PreMO or PostMO labels. For example, if the threshold is 0.5 and the probability of an observation being PostMO is greater than 0.5, the chunk would be assigned as PostMO. For Monkeys G and C, the best classifier performance was obtained with thresholds equal to 0.5 and 0.315, respectively, and signed distances to this movement onset threshold (MO threshold) were computed to estimate how far away a chunk's neural state was from the state of movement onset (see Materials and methods – Classification of PreMO and PostMO in NR and CO tasks). We found that *actual* PreMO and PostMO chunks exhibited significantly different distances to MO threshold for the held-out third of data from the CO task (*Figure 7a*, Blue: CO task, two-tailed Student's t-test on mean within-day predicted probabilities for held-out CO PreMO and PostMO, Monkey G: n = 6, T = −20.899, p=4.65e-06, Monkey C (inset): n = 4, T-4.2711, p=0.0236, Combined across monkeys: n = 10, T = −4.1097, p=2.638e-03). Thus, population spike count chunks reliably encode before and after movement onset in the CO task. Note that Monkey G exhibited about an order of magnitude more reliable separation between PreMO and PostMO than Monkey C (y axis in *Figure 7a*), and this is likely because of the lower neural signal quality in Monkey C (implanted ~3 years prior to study without resolvable single or multi-units) than Monkey G (implanted only ~1 year prior to study, with resolvable single and multi-units).

To assess whether the same spiking patterns were present during PreMO and PostMO in the NR task, the CO-trained classifier was used to predict the PreMO and PostMO labels of z-scored spiking activity chunks from NR trials. Note that in the NR task, chunks are labeled as 'PreMO' during the neurofeedback epoch of the NR task and before movement onset during the reaching epoch of the NR task, and labeled as 'PostMO' after movement onset during the reaching epoch. Given the differences in individual unit firing patterns across the CO and NR tasks (*Figure 6*), it is possible that the neural population activity also varies drastically across tasks and that the CO-trained classifier may not perform well when given NR population activity. Instead, we confirm that the same CO-trained classifier does yield significantly different distances to MO threshold for PreMO and PostMO chunks in the NR task (*Figure 7a*, Red: NR task, paired two-tailed Student's t-test on mean within-day probabilities for CO PreMO and PostMO chunks, Monkey G: n = 6, T = −7.459, p=6.83e-04, Monkey C (inset): n = 4, T = −13.591, p=8.62e-04, Combined across monkeys: T = −3.6193, p=5.578e-03, n = 10). This finding validates that the population reliably encodes gross kinematics similarly across tasks despite the individual unit activity changes observed across tasks in the previous analysis.

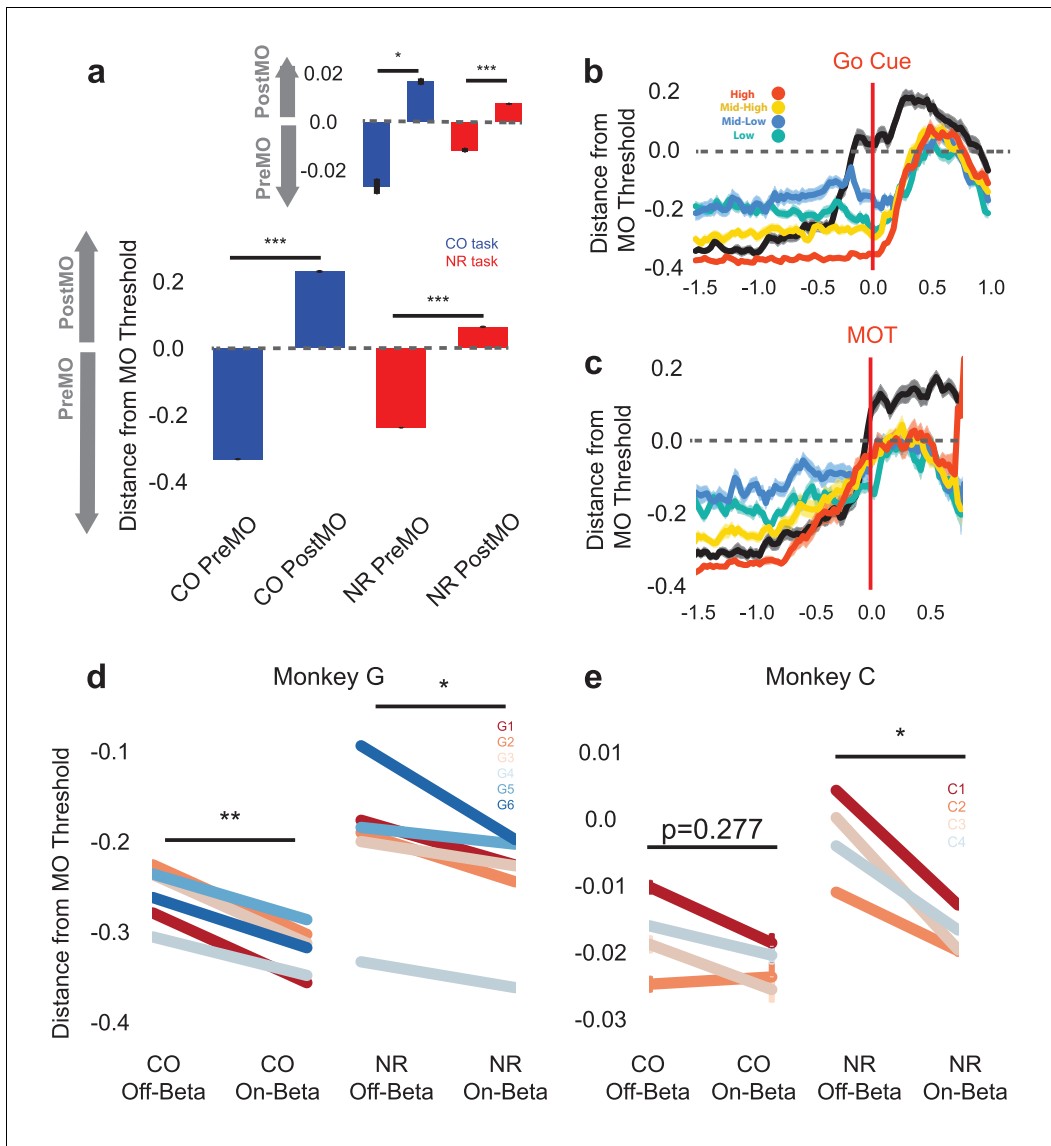

**Figure 7.** Population activity shifts away from MO threshold during on-beta timepoints in both the CO and NR tasks. (**a**) Distance to MO threshold for the CO (blue) and NR (red) tasks for PreMO and PostMO timepoints from Monkey G and Monkey C (inset). Bars less than and greater than zero indicate mean scores predicting PreMO and PostMO, respectively. (**b**) Example mean (s.e.m) of distance from MO threshold as a function of time to go cue for CO trials (black), high beta target NR trials (red), mid-high beta target NR trials (yellow), mid-low beta target NR trials (blue), and low beta target NR trials (green). At the go cue, distances to MO threshold are greatest for high beta target, and lowest for low beta targets. (**c**) Similar to (**b**) but aligned to MOT instead of go cue. Trials converge to MO threshold at MOT. (**d–e**) The mean distance from MO threshold for slow, preMO, off-beta and slow, preMO, on-beta timepoints during the CO (left) and NR (right) tasks for Monkey G and Monkey C, respectively. Individual lines connect mean off-beta and on-beta distances (s.e.m) for individual sessions.

The following source data and figure supplements are available for figure 7:

**Figure supplement 1.** Properties of Chosen vs Unchosen units and Chosen+ vs Chosen- units.

**Figure supplement 1—source data 1.** Sample sizes and Kruskal Wallis test results for *Figure 7—figure supplement 1*.

Given that the CO-trained movement onset classifier maintains its predictability in the NR task, we can now ask how the presence of beta oscillations changes the observed distance to the MO threshold. It is possible in both the CO task and the NR task that the presence of beta oscillations does not change the predicted distance to MO threshold as it is known that cells more involved in movement are no more likely to be entrained to beta oscillations than signals not involved in movement (*Murthy and Fetz, 1996a*). The classifier thus may have captured reliable movement signals from units that are unchanged when beta oscillations are generated, making the distance to MO threshold unaffected by the presence of the oscillations. A second possibility is that the presence of beta oscillations reflects neural activity that is further away from the MO threshold only during CO trials, but not NR trials. As the classifier weights were trained on the CO task, the classifier weights may reflect beta-related structure in the population that is useful for predicting CO bins to be PreMO. If the neurofeedback beta-related structure in the population is different from that in the CO task, as is suggested on an individual unit basis by *Figure 6*, then the population spiking patterns occurring during beta oscillations in the NR task may not exhibit similarly greater distances from the MO threshold. A final possibility is that despite the differences in individual unit firing patterns during the CO and NR beta oscillations, production of beta oscillations requires a consistent shift in population activity in both tasks, and that shift affects patterns predictive of movement onset.

By comparing the signed distance to the MO threshold during NR trials and CO trials during on-beta and off-beta timepoints, we can begin to discriminate amongst the three possibilities. *Figure 7b–c* shows the mean signed distance to MO threshold for NR trials of different beta targets and CO trials from one representative day (Monkey G, session 5). In *Figure 7b*, traces are aligned to the go cue, showing that during the neurofeedback epoch prior to the go cue, the high beta target (red) maintains a much greater distance from the MO threshold than the low and mid-low beta targets (green and blue). At the time of the go cue the high beta target is further from the MO threshold than the lower beta targets, suggesting that subjects must traverse a greater neural distance to arrive at the MO threshold, which may take longer resulting in a longer MOT. When aligning the same trials to the MOT, all traces converge around the MO threshold showing that all NR trials must arrive at the same MO threshold as CO trials in order to initiate movement. These examples suggest that the presence of more beta oscillations, as in the high beta target (red) trace, is an indicator of subjects' neural population being far from movement initiation.

It is possible, however, that the decoder has identified discriminative firing patterns that are unrelated to the presence of beta oscillations and that the differences observed in *Figure 7b–c* could be related to subjects' distinct beta-target strategy rather than the actual presence of beta oscillations. In *Figure 7b–c*, the green line corresponding to the lowest beta target is further from the blue line corresponding to the mid-low beta target, showing that the presence of more beta oscillations in the mid-low beta target is not the only factor in determining distance from the MO threshold.

To directly test whether the presence of beta oscillations affects distance to the MO threshold, we collapsed all NR data across beta targets, isolated slow chunks that occurred prior to *actual* movement onset, and compared distance to MO threshold for on-beta and off-beta bins in both the CO and NR tasks. *Figure 7d–e* shows mean on-beta and off-beta distances to MO threshold for individual days in the CO (left) and NR (right) tasks. In all cases but one, off-beta slow chunks exhibit significantly closer distances to MO threshold than on-beta slow chunks (paired Students' t-test of within-day means *Figure 7d*: Monkey G: CO off-beta vs. on-beta, n = 6, T = 6.742, p=1.089e-03, NR off-beta vs. on-beta, n = 6, T = 2.607, p=0.0478, *Figure 7e*: Monkey C: CO off-beta vs. on-beta, n = 4, T = 1.325, p=0.277 NR off-beta vs. on-beta n = 4, T = 4.763, p=0.0176, Combined across monkeys: CO off-beta vs. on-beta, n = 10, T = 4.246, p=2.160e-03, NF off-beta vs. on-beta, n = 10, T = 3.185, p=0.0111). The CO and NR tasks exhibit common population-level activity changes reflected by the onset of beta oscillations, and specifically, these population changes encode a shift further away from the MO threshold.

## Properties of units contributing most to PreMO versus PostMO classification

To determine the properties of units that contributed most to the PreMO and PostMO predictions of the logistic classifier, we chose to analyze units that fulfilled two criteria. The first criterion was that a logistic classifier trained only on CO chunks from the individual unit, not the entire population

as in *Figure 7*, had to predict significantly lower scores for PreMO than PostMO in held-out data in the CO and NF tasks. The second criterion was that units had to exhibit lower predicted scores for on-beta, slow chunks than off-beta, slow chunks in both tasks. The units that fulfilled both criteria were referred to as 'chosen' units, and were compared with all other 'unchosen' units (units collapsed across days – Monkey G: 104 chosen units, 251 unchosen units, Monkey C: 171 chosen units, 342 unchosen units). We analyzed the differences between chosen and unchosen weights using five different metrics—classifier weight, beta amplitude-to-spike rate slope during CO task, mean modulation during the CO task, mean firing rate during the CO task, and beta rhythmicity during the CO task (see *Supplementary file 1* – Chosen versus Unchosen Unit Analysis, *Figure 7—figure supplement 1*, and *Supplementary file 1* – Chosen versus Unchosen Units).

We found that the weights of chosen units exhibited significantly higher and more positive weights than unchosen units, indicating that most chosen units increased their firing rate during PostMO compared with PreMO. Chosen units also tended to have higher firing rates. Further, chosen units showed significantly lower beta amplitude-to-spike rate slopes during the CO task than unchosen units, consistent with the finding that firing rates increase during movement concomitantly with beta amplitude decreases. Chosen units and unchosen units showed no difference in their mean task modulation, and no difference in their beta rhythmicity (*Figure 7—figure supplement 1*, and *Supplementary file 1*).

Lastly, to assess any difference between the chosen units with a positive classifier weight and the chosen units with negative classifier weight, the same analyses were performed between these groups (termed 'Chosen+' units and 'Chosen–' units, Monkey G: 85 Chosen+ units, 19 Chosen– units, Monkey C: 170 Chosen+ units, 1 Chosen– unit). The Chosen+ units and Chosen- units, respectively, increase and decrease firing rate during movement. Thus, we also found that the Chosen+ group exhibited significantly lower beta-to-firing rate slopes. The groups did not show any difference in mean task modulation. The positive units exhibited significantly higher firing rate. Finally, the negatively modulated units had a non-significantly higher 'beta rhythmicity' than the positive units.

Overall, units that contributed most to the classification of PreMO versus PostMO tended to be high firing units that were positively modulated with movement onset, yet were not necessaily units that were more modulated during the CO task. Within this group were a few units that fired less with movement, that were generally lower firing rate, and possibly more entrained to ongoing beta oscillations in the local field potential. These Chosen- units may be a distinct subpopulation of single and multi-units that act as pacemaker cells for the population (*Chen and Fetz, 2005*), although more data and a more careful characterization of firing properties, ISIs, and waveforms is needed to make this claim.

## Discussion

We have provided evidence that volitionally increasing and decreasing beta power in the motor system with neurofeedback achieves neural states that precede slower and faster movement onset times, respectively, in three monkeys. These results support the hypothesis that beta oscillations in the motor system reflect neural patterns that are far from a movement onset neural state. Importantly, we used simultaneously recorded single- and multi-unit activity during the NR and CO tasks to characterize how the presence of beta oscillations reflected changes in underlying neural population activity, which ultimately drives the behavioral changes observed. During the neurofeedback epoch of the NR task, population neural activity exhibited greater distances from the computed MO threshold when beta oscillations were observed, and shorter distances when there were no oscillations. We emphasize that this result is not merely driven by the observation that there are often more beta oscillations at rest than during movement. Rather, when subjects are performing neurofeedback and they are at rest, their underlying neural population is shifting further away from the MO threshold when beta oscillations are observed.

Next we discuss how our results mesh with other reported behavioral correlates of beta oscillations, putative mechanisms of how beta oscillations are generated and may affect movement generation, and benefits of using neurofeedback as an experimental paradigm.

## Beta-generating mechanisms in the motor system resolve conflicting reports of beta-behavioral correlates

Our result that volitionally increasing and decreasing motor cortical beta power with neurofeedback precedes slower and faster movement onset times supports the hypothesis that beta oscillations are linked to neural patterns that slow onset of new movements (*Gilbertson et al., 2005*). Our results add to previously reported findings of elevated beta power prior to and after well-trained movements (*Donoghue et al., 1998*; *Murthy and Fetz, 1992*; *Sanes and Donoghue, 1993*). Although we cannot identify the mechanism that drives the beta oscillations observed, modeling and *in vitro* slice work shed some light. Recent modeling of striatal neural populations showed that increased medium spiny neuron (MSN) excitation can result in beta oscillations within the striatum, which can propagate through output structures of the basal ganglia (*McCarthy et al., 2011*). MSN excitability is affected by neuromodulators including acetylcholine (*Cannon et al., 2014*), which notably drives increased MSN excitability primarily in D2 MSNs, or the MSNs responsible for the indirect pathway activation (*Benarroch, 2012*). The findings from our study show that when cells generate beta oscillations they encode a slower movement state, which could reflect indirect pathway activation. Recent work has also suggested that shifts in attention caused by salient stimuli involve the intralaminar nuclei of the thalamus (*Smith et al., 2009*), which projects to the striatum (*Smith et al., 2004*), potentially also resulting in transient increases in beta power in D2 MSNs (*McCarthy et al., 2011*). This common striatal beta-generating mechanism could explain how increases in attention have been reported to evoke beta oscillations (*Fetz, 2013*; *Leventhal et al., 2012*; *Saleh et al., 2010*), and could be used to pause current motor programs in response to salient stimuli (*Benarroch, 2012*). This mechanism is also a plausible explanation for evidence of beta oscillations occurring during untrained, free-reaching movements (*Donoghue et al., 1998*; *Murthy and Fetz, 1996b*). These oscillations could be driven by salient stimuli that subjects encounter as they execute and update their internally generated motor plan.

Our results and proposed mechanism of beta generation do not predict oscillatory events occurring during isometric contraction (*Baker et al., 1997*; *Conway et al., 1995*); however, it is becoming increasingly common to find different mechanisms for generating similar frequency oscillations (*Cannon et al., 2014*). In vitro slice work has identified that with sufficient excitatory drive to slices of sensorimotor or motor cortex, beta frequency oscillations emerge in deep cortical layers (*Roopun et al., 2006*; *Yamawaki et al., 2008*). It is possible that the strong excitatory drive needed to stiffen muscles during an isometric contraction task, in contrast to reaching movements that require temporal coordination of antagonist muscle groups (*Georgopoulos, 1992*), is sufficient to generate beta oscillations by the same means described by *Roopun et al. (2006)* and *Yamawaki et al. (2008)*. Further evidence for this proposed mechanism comes from computational models of driving motoneuron recruitment with pyramidal tract neurons. When pyramidal tract neurons fire at beta band or higher frequencies, motoneurons increase recruitment and hence muscular force production (*Watanabe and Kohn, 2015*), as required in an isometric contraction task. Potentially, beta oscillations are observed during isometric contraction tasks because of large muscular force requirements in the task, not because of the same striatal beta-generation pathway previously described.

Finally, our results and the proposed striatal beta-generating mechanism do not predict the correlation of beta power with other behavioral metrics such as movement onset speed, peak speed, and onset acceleration, which were correlated with beta power in other studies (*Joundi et al., 2012*; *Pogosyan et al., 2009*). Note though, that the findings (*Joundi et al., 2012*; *Pogosyan et al., 2009*) were from experiments using transcranial alternating current stimulation (tACS) at beta frequencies applied to motor cortical areas. The mechanisms of tACS are still unclear (*Zaghi et al., 2010*), so it is possible that the reported behavioral effects result from evoked neuronal activity patterns that are specific to tACS stimulation and do not occur endogenously.

Although identifying mechanisms that generate beta oscillations can shed light on how certain types of behavior such as attention or isometric contraction may be correlated with onset of beta oscillations, they do not provide information on how the dynamics of the underlying neural population generate movement change.

## Encoding shifts in population neural activity related to beta oscillations

When populations of neurons generate beta oscillations, their patterns are further from movement onset threshold than when they do not generate oscillations. In both the CO and NR tasks, on-beta PreMO bins exhibit further distances from the MO threshold than off-beta PreMO bins (*Figure 7*), emphasizing the common population shift that occurs in the CO and NR tasks during beta oscillatory periods. We propose that neural populations must stop generating beta oscillations before they can initiate specific preparatory and movement generating patterns that may occur closer to the MO threshold. Thus, the generation of beta oscillations favors a low-risk state in which neural populations will not accidentally create patterns that cause movement, and in exchange compromise their readiness for upcoming movements. Generation of these oscillations may possibly be implemented by a distinct subpopulation of 'pacemaker' cells (*Figure 7—figure supplement 1*), although data with more clearly isolatable single units are needed to describe how distinct subpopulations may each contribute to the pacing of beta oscillations versus the encoding of kinematic information.

The hypothesis that encoding of specific movements is compromised with beta oscillations is further supported by experiments showing that in a delayed reaching task similar to the CO task here, movement cues associated with more uncertainty are correlated with higher beta power during preparatory periods than movement cues associated with certainty (*Tzagarakis et al., 2010*). Uncertain stimuli bias the subject against preparing movements, corroborating that periods of beta oscillations are associated with compromised preparation. Further, during spike-driven cursor brain-machine interface tasks in which subjects are not moving, periods of beta oscillatory activity correspond to inferior neural decoding (*Milekovic et al., 2013*) and slower cursor movements (*Ranade et al., 2009*), suggesting that the population contains less specific directional information that can be used to move the prosthetic cursor.

How does the hypothesis of beta oscillations reflecting less informative encoding mesh with results showing linearly separable neural activity patterns during preparatory periods, when beta oscillations are prominent, for different upcoming movements (*Hatsopoulos et al., 2004*)? As beta oscillations occur in transient bursts and only show elevated power for trial averages (*Sherman et al., 2016*), it is possible that movement encoding during bursts is compromised, but outside of bursts is intact. Further, not all cells engage in beta oscillatory events (*Murthy and Fetz, 1996a*), making it possible that some cells are still encoding movement information during ongoing oscillations (*Reimer and Hatsopoulos, 2010*).

## Neural activity in Parkinson's disease (PD) reflects slowed, non-specific kinematic encoding and elevated beta synchrony

Individual motor cortical neuron recordings from parkinsonian non-human primates and electrocorticography recordings in humans support the hypothesis that beta oscillations, found to be excessive in PD throughout the cortico-basal ganglia-thalamic loop (*Cannon et al., 2014*), compromise the encoding of kinematic information needed for movement generation. Indeed motor cortical cell activity is more correlated in motor disabled non-human primates after systemic treatment with 1-methyl-4-phenyl-1,2,3,6-tetrahydropyridine (MPTP) than before treatment (*Goldberg et al., 2002*; *Pasquereau and Turner, 2011*). Cells also fire less specifically to passive limb movements after MPTP treatment (*Goldberg et al., 2002*), suggesting a correlation between excessive beta oscillations and reduced specificity for movements. Also, it was found that coupling between low frequency beta phase and high frequency broadband gamma power was elevated in PD patients who had their deep brain stimulation therapy turned off (*de Hemptinne et al., 2015*) or were in an off-state following an extended period of no levodopa medication (*Swann et al., 2015*). Both stimulation and levodopa administration therapies improve symptoms of bradykinesia and rigidity for PD patients, suggesting that synchronization of high gamma, a proxy for spiking activity, to beta frequencies may be related to akinesia symptoms. This evidence, although in a disease model, supports the hypothesis that beta oscillations reflect reduced capability of kinematic encoding in population neural activity.

## Advantages of using neurofeedback to study behavioral correlates

Using a neurofeedback paradigm to investigate behavioral and population neural correlates of oscillations has several advantages. First, as the neurofeedback epoch only requires subjects to modulate

beta power and to be seated at rest, they choose their own subject-specific strategy for generating or quenching beta activity. These strategies may include co-modulating other frequency bands, imagining movement, or performing other internal behaviors that generate beta activity. For example, while Monkeys C and G inversely modulated low frequency (1–10 Hz) power with beta frequency power, Monkey S did not modulate 1–10 Hz power as drastically but did exhibit increased 50–60 Hz power with increased beta power (*Figure 2d–f*). Despite the different approaches that were taken to increase and decrease beta power across animals there is still a consistent effect of high versus low beta power on movement onset times, increasing confidence that the oscillation is a reliable marker of the observed behavior. Many motor tasks previously used to study beta oscillations engage motor preparation, increased attention, cue expectation, and possibly muscular stiffening simultaneously within the task. These overlapping behaviors make it challenging to correlate a specific behavior with the presence of beta oscillations. Using a neurofeedback task, in which there will be individual subject-specific neurofeedback strategies, prior to a single task allows for a better test of how tightly linked the conditioned neural feature is to behavior.

Another advantage of using neurofeedback over other approaches to perturb neural oscillations such as non-invasive transcranial alternating current stimulation (*Joundi et al., 2012*; *Pogosyan et al., 2009*) or invasive transcranial electrical stimulation (*Ozen et al., 2010*), is that the recorded neural signal is not tarnished with a stimulation artifact. In this study, simultaneously recorded units were analyzed and shown to exhibit different activity during neurofeedback-induced beta oscillations compared with natural beta oscillations occurring during typical reaching tasks. Despite differences at an individual unit level, population-level analyses show that beta oscillations promote a consistent movement-slowing state during both the CO and NR tasks, matching what is observed behaviorally. This analysis could be compromised if a stimulation artifact prevented recording of local field potentials or single units.

Another report, *McFarland et al. (2015)*, using a similar but non-invasive beta band neurofeedback method prior to a movement task reports comparable behavioral results. The authors found that three of the eight subjects exhibited significant reductions in movement onset time following reduced beta power, consistent with our findings. It is possible that the movement onset increase was not seen for all subjects because there was substantially more temporal smoothing in the study neurofeedback task setup (neural signals averaged in window of 1 s, compared with our window of 200 ms), and no hold requirement for the neurofeedback cursor (compared with our hold requirement of 450 ms). Thus, their subjects could be in a greater range of neural states prior to the movement task, making the movement onset versus beta target relationship less robust. Finally, the authors do report a group-level significant increase in movement accuracy following beta reduction, a metric that did not change in our experiment likely because of the subjects' overtraining of arm reaches in our study.

Finally, neurofeedback is a tool that, if effective at introducing a change in subsequent behaviors, could be a directly translatable therapy for patients. For example, if excessive synchronization of motor neurons is pathological in PD, learning to reduce neural coupling with neurofeedback of beta oscillations may improve bradykinesia symptoms. Evidence suggests that PD patients do indeed exhibit stronger movement-related beta power reduction prior to movement onset than non-PD patients (*Rowland et al., 2015*), implying that some patients may already reduce beta power to initiate movement more easily. Thus, neurofeedback could be a tool to teach patients to cognitively modulate their beta power for symptom improvement (*Khanna et al., 2016*; *Moxon and Foffani, 2015*).

## Alternative explanations

An alternative explanation to our results is that the monkeys used movement to decrease the beta cursor and/or isometric contraction to increase the beta cursor during the neurofeedback epoch, and it was this movement that drove the MOT differences. First, a previous group has published work with NHPs performing beta-range neurofeedback (30–43 Hz instead of our 25–40 Hz) and reported a lack of overt movement evidenced by video and EMG recordings (*Engelhard et al., 2013*). Second, in our studies, we took video of Monkey C and Monkey G performing the task and found no overt movements when subjects were performing neurofeedback. Third, if the monkeys did make voluntary, overt movements, they would likely be moving their left hand or legs to assist with desynchronization of beta power during the low beta target trials of the neurofeedback epoch.

Evidence from human psychophysical studies supports the finding that limb movements contralateral to the hand performing a simple reaction time task actually lengthen subjects' reaction time (*Begeman et al., 2007*; *Buenaventura and Sarkin, 1996*). Thus, measured right hand MOTs in this study would be slower if monkeys were moving their left hand or leg. Finally, subjects could be using isometric contraction to increase beta power to assist in getting to the high beta target, as demonstrated in *Baker et al., 1997*. However, a previous study showed that if the hand of interest (here, right hand) is isometrically contracted, there is no significant slowing or quickening in MOT compared with a subject performing a simple reaction time test at rest (*Castellote et al., 2004*). Thus, previously published work and our own video evidence suggests that subjects are not moving. Even if subjects were moving to lower the beta cursor, human psychophysics studies suggest that MOT should be slower than if they were not moving. Thus, moving would make our MOT effect less significant, not more significant, alleviating the concern that monkeys moving is causing our reported MOT trend.

Another interpretation of our results is that as subjects only perform reaches to well-practiced, predictable targets, the finding that the presence of beta oscillations correlates with slowed MOTs does not apply to the general case where movement intentions are not over-practiced. Unfortunately, two challenges exist to testing the effect of increasing or decreasing beta oscillations on MOTs in unprepared arm-reaches in our task setup. First, unprepared reaches have greater MOTs (*Churchland et al., 2006*) and the beta state reached by the neurofeedback manipulation does not persist for long (*Figure 2a–c*). The longer it takes to process an unprepared reach cue, the further away in time this will make subjects' MOT from the end of the neurofeedback epoch, making it likely that by the time MO occurs, the effect of the neurofeedback will have washed out. Second, salient cues are known to evoke beta oscillations (*Saleh et al., 2010*), making any post-neurofeedback cue indicating reach target likely to equalize subjects' beta states. For both these reasons, we were unable to test the hypothesis that the presence of beta oscillations has a direct effect on unprepared movements. However, previous evidence showing the emergence of beta oscillations with salient and used cues (*Leventhal et al., 2012*; *Saleh et al., 2010*), and the emergence of higher beta power with cues associated with less certainty (*Tzagarakis et al., 2010*), suggest that beta oscillations may also reflect the neural states that subjects are in as they process cues to prepare their next movements.

## Summary

The results from this study show that beta band frequencies are a marker of slowed movement onset. The underlying motor cortical population activity shifts further from a MO threshold when subjects volitionally produce beta oscillations in the NR task or naturally produce them in the CO task. We discussed mechanistic generators of beta oscillations and used these to reconcile discrepancies in previously reported behavioral correlates of beta activity. Finally, we proposed that the shifts observed in activity of motor cortical neurons caused by generation of beta oscillations reduce the probability of accidental movement in exchange for compromised encoding of specific movement plans.

# Materials and methods

## Surgery, electrophysiology, and experimental setup

Three male rhesus macaques (*Macaca mulatta*, RRID: NCBITaxon:9544) were chronically implanted with arrays of 128 Teflon-coated tungsten microwire electrodes (35 µm in diameter, 500 µm separation between microwires, 16 × 8 configuration, 6.5 mm length, Innovative Neurophysiology, Durham, NC) in the left upper arm area of primary motor cortex (M1) and posterior dorsal premotor cortex (PMd). Localization of target areas was performed using stereotactic coordinates from a neuroanatomical atlas of the rhesus brain (*Paxinos et al., 2013*). LFP activity was recorded at 1 kHz using either the 128-channel Multichannel Acquisition Processor (Plexon, Inc., Dallas, TX) (Monkeys S, G) or the 256-channel Omniplex D Neural Acquisition System (Plexon, Inc.) (Monkey C). Single-unit and multi-unit activity from Monkey G was manually sorted offline using Offline Sorter (Plexon, Inc). Channel-level activity (*Chestek et al., 2011*) from Monkey C was defined using OmniPlex's auto-threshold procedure to set each channel threshold to 5.5-standard deviations from the mean

signal amplitude. Thresholds were set at the beginning of each session based on 1–2 min of neural activity recorded as the animal sat quietly (i.e. not performing a behavioral task).

Monkeys S and G were trained to perform a center-out delayed reaching task using a KINARM exoskeleton (BKIN Technologies, Kingston, ON, Canada) fitted to their right arm. Monkey C was trained using a custom right-arm sleeve with a red LED marker on the hand that was tracked in real time with an Impulse X2 motion capture system (PhaseSpace, San Leandro, CA). For all monkeys and tasks in this study, visual feedback of hand position was shown by a circular cursor on the task screen. Monkeys S and G right arm movements were restricted to the horizontal plane by the KIN-ARM. Monkey C could rest and move his right arm on a horizontal plane like Monkeys G and S, but could also move his arm above the plane. Prior to this study, Monkey S was trained at reaching tasks and spike-based brain-machine interface (BMI) cursor tasks for 4 years, Monkey G was trained at joystick tasks and spike-based BMI cursor tasks for 1 year, and Monkey C was trained at reaching and spike-based BMI cursor and virtual exoskeleton tasks for 3 years. All procedures were conducted in compliance with the NIH Guide for the Care and Use of Laboratory Animals and were approved by the University of California, Berkeley Institutional Animal Care and Use Committee.

## Center-out reaching task

Subjects performed a center-out (CO) reaching task consisting of right hand movements from a center target to a peripheral target distributed over a 13 cm diameter circle (*Figure 1a*). The workspace was created to minimize any requirement for postural changes during task performance. Target radius was typically 1.2 cm in the workspace. Trials were initiated by entering the center target and holding for a variable time (uniformly distributed within 200–800 ms). The go cue after the hold period was indicated by the center target changing color and the peripheral target illuminating, cuing a reach to that target. A liquid reward was provided after a successful reach to each target and a peripheral hold period of 200 ms.

## Neurofeedback-reaching task

In the neurofeedback-reaching (NR) task monkeys performed neurofeedback immediately followed by an arm reach to a peripheral target (*Figure 1b,c*). To initiate a trial, subjects moved their right hand to a central target and held for 200 ms after which the neurofeedback epoch began. Throughout the neurofeedback epoch the right hand was required to remain in the center target otherwise a timeout ensued. A yellow square beta cursor representing an estimate of the monkey's motor cortical beta power and one of four beta targets appeared on the left of the screen. As estimates of normalized motor cortical beta power increased or decreased, the beta cursor position increased or decreased on the y axis, updating every 100 ms. Subjects modulated their endogenous neural signals to make the beta cursor hover over the beta target for 450 ms to successfully end the neurofeedback epoch. Subjects had 60 s to reach the beta target with their beta cursor before the trial timed out and the same beta target repeated. Once the neurofeedback epoch was completed the reaching epoch began. The beta cursor and beta target disappeared and a peripheral reach target appeared in a location 6.5 cm away from the center target. The peripheral target was kept in a consistent location from trial to trial and only changed on a day-to-day basis, making its location predictable. Once subjects reached the peripheral target and held for 200 ms, they received a liquid reward. No reward was given for trials in which the subject did not successfully complete the neurofeedback epoch within 60 s (neurofeedback timeout error), hold in the peripheral reach target for the full 200 ms (hold errors), or in which the subject did not reach the peripheral target within 5 s (timeout error).

## Calculation of beta neurofeedback cursor

The beta neurofeedback cursor was designed to reflect endogenous beta oscillatory events in motor cortex. To calculate its position at each time point (every 100 ms), LFP signals from three channels in the most anterior part of the implanted multi-electrode array were manually selected for each subject. Minimal artifacts and strong beta oscillatory activity during the reaching only task were the selection criterion, as in *Engelhard et al. (2013)*. These channels were kept consistent throughout all NR tasks for that subject. Subjects' LFP signals were streamed over a local intranet via the PLEXNET client-server application (Plexon Inc.). In the task application, the most recent 200 ms of neural data

were used to estimate the beta power (25–40 Hz) and broadband power (1–100 Hz) using the multi-taper method (*Babadi and Brown, 2014*; *So et al., 2014*) with five tapers. Beta band and broadband estimates were then averaged across channels, and finally a normalized estimate of beta power ($\beta_{est}$) was computed by dividing the beta band power estimate by the broadband power estimate:

$$\beta_{est} = \frac{\frac{1}{3}\sum_{ch=1}^{3}\ \sum_{f=25\,Hz}^{40\,Hz}\ PSD_f^{ch}}{\frac{1}{3}\sum_{ch=1}^{3}\ \sum_{f=1\,Hz}^{100\,Hz}\ PSD_f^{ch}}$$

Once $\beta_{est}$ was computed, a subject-specific linear transform was used to map $\beta_{est}$ to a vertical screen position. A two-timestep (200 ms) moving average (boxcar) filter was then used to smooth out the displayed beta cursor position.

*Table 1* lists the normalized beta values needed to modulate to the middle of each beta target for each monkey. These values were finalized after ~1 week of training the subjects on the beta neurofeedback task. Initial training began with more lenient beta neurofeedback requirements (values to achieve low beta and high beta target were closer to the mean beta cursor value). As subjects improved in performance, the beta targets moved further apart until the top and bottom targets had a mean time to target of 5–10 s.

## NR task variant using beta band and 1–10 Hz

Monkey G performed one task variant of the NR task where the neurofeedback cursor in the x-axis was controlled with normalized 1–10 Hz power in addition to it being controlled in the y-axis with normalized beta power, as described above. The four beta targets were in the same locations as in the standard NR task. This variant thus required that the subject produce the correct normalized beta power value without simultaneously producing too high or too low a 1–10 Hz power value, as extreme 1–10 Hz modulations would move the cursor too far to the left or to the right of the beta target. The subject practiced the task with lenient requirements at first which were slowly made harder until performance plateaued and the mean time to LFP target was 5–10 s. The final values that normalized 1–10 Hz power must fall within were 0.74–0.95.

## Chance level performance of beta neurofeedback epoch

For each session (typically lasting 10–40 min), beta cursor position from the session's online performance was re-run through a target-shuffled task simulation, designed to assess whether the subjects' performance during the beta neurofeedback epoch of the NR task was resulted merely from chance fluctuations in beta power or was caused by volitional changes in beta power that were specific for the beta target on the screen. In the simulation, after the beta cursor entered the beta target and held for the 450 ms beta target hold time, an average arm-reaching time, the constant reward time, and the constant inter-trial interval time transpired to simulate the natural pacing of the task. At the end of the simulation, a metric of chance performance was the mean number of successful beta targets acquired over the length of the session. For example, if one target-shuffled performance yields 10 successful trials in 10 min, the chance rate for that simulation would be one rewarded target per minute.

One hundred simulations were run per session (each session ~10–40 min) yielding a distribution of rewarded targets/minute. The mean and standard deviation of the distribution was calculated, and used to z-score the actual number of rewarded trials. The resultant z-scores for each session are plotted in *Figure 1e*, where each point corresponds to a session (session $i$), and each point's position on the x axis ($x_i$) corresponds to the first trial in session $i$.

## Power spectral densities (PSDs)

In *Figure 1d*, a movement-onset aligned, detrended spectrogram (window size is 200 ms, step size is 10 ms, multi-taper method with five tapers) is plotted from 1 s before movement onset to 0.75 s after movement onset. The spectrogram estimate was detrended by first estimating power as a linear function of frequency (power = $m$*frequency + $b$) using all trials and timepoints as data to fit $m$, $b$ in the linear regression. Then, the estimated power from the linear regression was subtracted from the computed spectrogram, effectively flattening the typical 1/f trend exhibited by neurological signals. The detrended spectrogram was averaged over trials, and is plotted in *Figure 1d*.

In *Figure 2a–c*, normalized beta power traces are shown for the last 1000 ms of the neurofeedback epoch through the first 500 ms of the reaching epoch. These are computed the same way as detailed above in 'calculation of beta neurofeedback cursor', except that they are not transformed from normalized beta power to the cursor position and they are calculated at time steps of 10 ms instead of 100 ms. The window width and spectral computation method is the same.

*Figure 2d–f* shows the mean full power PSDs for the last 800 ms of the neurofeedback epoch to the first 200 ms of the reaching epoch for each beta target. This window was selected because first, it overlapped with the beta target hold period, a period in which beta power values must be specific for each target. Second, it was a long enough window to accurately capture low frequency activity. To compute the values at each frequency for each of the beta targets, first a PSD is computed for all trials yielding matrix $X$ of size trials ($n$) x frequencies ($f$). The mean and standard deviation is computed for each frequency, yielding vectors $M$ and $S$, both of size $f$ x 1. The matrix $X\_zscore$ is computed by subtracting off $M$ from each row of $X$, and then dividing each row by $S$. Finally, trials to each beta target are averaged together and plotted in *Figure 2d–f*.

## Calculation of on-beta and off-beta labels

In the text we compared what percentage of time points are part of a beta oscillatory event (on-beta) for each beta target. We also used the concept of on-beta and off-beta time bins in *Figures 5j*, *6a–b* and *7d–e*. To determine when the beta oscillatory events occurred, the following procedure was applied. The local field potential (LFP) signal from one of the electrodes used in the neurofeedback task was selected, and filtered at 25–40 Hz with a fifth order Butterworth filter. The amplitude of the filtered signal was estimated with the Hilbert transform. Each day, a distribution of amplitude values was computed by aggregating all rewarded trials from that day. A threshold was set at the 60th percentile of the resultant distribution. Then, all timepoints surpassing the threshold were labeled with a 1. All other points were labeled with a 0. Then, the length of each block of uninterrupted '1s' was computed, and if the block was longer than 125 ms (so the oscillatory episode lasted at least 3–4 cycles as described in *Murthy and Fetz (1996a)*; *Sherman et al., 2016*), the block remained as is. If the block was less than 125 ms, all points in the block were changed back to '0s'. All of the above computations were done at the original LFP sampling rate of 1000 Hz. For analyses using bins longer than 1 ms, each bin was labeled as a '1' or '0' depending on what the majority of points within the bin were labeled as.

## Calculation of slow versus fast timepoints

Throughout the study slow and fast timepoints are distinguished. A threshold of 3.5 cm/sec applied to hand speed was used to separate slow from fast. This threshold was chosen because using it yielded the best discrete classifier performance separating slow spiking patterns from fast spiking patterns in Monkeys G and C's CO task in comparison with other thresholds (data not shown).

## Movement onset time calculation

To compute movement onset time for the arm-reaching epoch in the NR task, we used a method inspired by *Churchland and Shenoy (2007)*. First, the x and y cursor velocity over the course of the reach (t x 2) was projected onto a unit vector pointing from the center to peripheral target. The norm of the projected cursor velocities yielded a single time series of projected speed (t x 1). The index and value of maximum projected hand speed was computed and called ($i$, $M$). Starting at $i$, we scanned backwards in time ($i$-1, $i$-2, . . .) until the value of the projected hand speed fell below 20% of the peak hand speed ($0.2*M$, see *Figure 1f*). This timepoint was called the movement onset time and was used because it allowed for comparisons across different reach targets, animals, and experimental setups without having to define absolute hand velocity cutoffs. Example hand speed traces and marked movement onset times are shown in *Figure 1f* (mean projected hand speed traces and mean movement onset time) and *Figure 1g* (individual trial projected hand speed traces and marked movement onset times in black dots).

## Ordering statistics

Cuzick's test for trend (*Cuzick, 1985*) is a non-parametric test for significant ordering of groups in an increasing or decreasing manner (two-tailed), and was used to assess significance of ordering of

behavioral metrics according to the four beta target. A test statistic (Z) is calculated for the hypothesis that groups follow a designated ordering (*Cuzick, 1985*). Z was calculated using the ranks of individual points and the group assignment (assignments used here: low beta target: 1, mid-low beta target: 2, mid-high beta target: 3, high beta target: 4 except for *Figure 5g*, right where the assignments were: low beta target: 4, mid-low beta target: 1, mid-high beta target: 2, high beta target: 3), to determine whether there is a significantly increasing or decreasing metric following the group ordering. Z follows a standard normal distribution (confirmed for data here by shuffling group labels 10,000 times and comparing the resultant Z distribution to a standard normal distribution with the KS test), so a p-value can be calculated using the cumulative standard normal distribution.

## Trial re-allocation analysis

In *Figure 5*, trials are relabeled by spectral features to simulate the movement onset time analysis in *Figure 3* as though another frequency band or beta power calculation method was being used to control the neurofeedback cursor. Each subplot of *Figure 5a–f* contains all *n* trials from that subject's NR task performance (except for a rejection of trials with movement onset times outside of the range of 0.0–0.7 s). Here, trials are assigned group labels according to which quartile their power estimate for a particular frequency band falls in.

For *Figure 5a–c*, non-normalized beta is computed in a window including the last 800 ms of the neurofeedback epoch. Trials were assigned labels depending on where they fell in the distribution of non-normalized beta power across trials: 0-25th percentile: label: 1, 25th – 50th percentile: label: 2, 50th – 75th percentile: label: 3, and 75th – 100th percentile: label: 4. Then, Cuzick's test was performed using the newly assigned labels to assess whether MOTs for non-normalized beta power exhibit the same significant ordering.

For *Figure 5d–f*, normalized non-beta frequencies were used to re-assign labels, and Cuzick's test was again run.

## Individual unit analysis

All population unit analysis was only conducted with Monkeys G and C. Monkey S had arrays that had been implanted for >3 years and were no longer usable for recording high frequency activity. All neural data from sessions from Monkey G were offline sorted using the Plexon Offline Sorter. Isolated single units and multi units were included in analysis. For Monkey C, channel activity (*Chestek et al., 2011*) was used (see Surgery, electrophysiology, and experimental setup). Analyses were performed within a day to prevent day-to-day recording instability from influencing analysis.

## Slow, on-beta mean firing rate comparisons

To assess whether units exhibited the same mean firing rate during neurofeedback-induced or naturally occurring beta oscillations, spiking activity was binned into 100 ms bins during NR trials and CO reaching trials. Spike counts during the slow, on-beta bins in the neurofeedback epoch of the NR task were compared with spike counts during slow, on-beta time points during the CO task. A Mann–Whitney test was used to assess significant differences in the distribution of firing rate for each unit across the two tasks. The percentage of cells exhibiting significant differences is shown in *Figure 6b*.

## Beta amplitude-to-spike rate mapping

To assess how individual units were influenced by ongoing beta oscillations in different task contexts, a beta amplitude-to-spike rate mapping was estimated for each unit in the two different tasks. The methods described below were adapted from *Canolty et al. (2012)*. See the original work for more methodological details. To compute the beta amplitude-to-spike rate mapping, rewarded CO and NR task trials were extracted from 1.5 s prior to the reaching go-cue to 1.0 s after the reaching go cue.

For these trial time slices, spiking data from all units were extracted and binned into 1 ms bins yielding vectors of length 2500. These were concatenated yielding a long (ntrials*2500 x nunits) matrix of spike counts for each task. Continuous beta amplitude was estimated for these time slices using a fifth order Butterworth bandpass filter from 25 to 40 Hz and then applying the Hilbert transform. These vectors were then concatenated yielding a long (ntrials*2500 $\times$ 1) vector of beta

amplitudes, one for each task. The beta amplitude vectors were both reordered by increasing amplitude, and the spiking data were reordered to keep the corresponding time points aligned. Each reordered beta amplitude vector was then reduced to 10 values, where each value is the mean of 1/10th of the sorted beta amplitude array (mean of first 10th of array is first value, mean of second tenth of array is second value, etc.). Mean spike rate values were estimated in the same way, yielding a 10 × 1 beta amplitude and a 10 x nunits spike rate array for both the CO and NR tasks. Plotted in *Figure 6c–e* are examples of CO (blue) and NR (red) mappings from log10(beta amplitude) to spike rate. A linear regression is fit using the 10 ordered pairs and plotted.

To assess the intra-task and inter-task stability of the beta amplitude to spike rate mapping, in *Figure 6f and g*, two equal, non-overlapping segments of data (set 1: indices [0, 2, 4, 6...] set 2: indices [1, 3, 5, 7...] in the (ntrials*2500 x 1) vector of beta amplitudes and (ntrials*2500 x nunits) matrix of spike counts) were used to estimate the beta amplitude to spike rate mapping. The slopes from the resultant linear regressions were plotted against one another. If the mapping estimate is stable, the slope acquired from slice 1 of the data ought to match the slope acquired from slice 2 of the data.

For Monkey C, there was very limited CO task data, so fitting the beta amplitude-to-spike rate mapping with a subset yielded unstable inter-task correlations. Using 75% of data in each subset yielded more stable correlations and was used in *Figure 6f–h* and subsequent statistics.

## Classification of PreMO and PostMO in NR and CO tasks

To assess the information about movement onset contained in spiking activity during different parts of the NR and CO tasks, a logistic regression classifier was trained on spiking activity to discriminate pre-movement onset (PreMO) and post-movement onset (PostMO). First, trials were extracted from 1.5 s prior to the go cue to 1.0 s after the go cue yielding 2.5-second-long trials. Spiking activity from all CO reaching sessions and all NR task sessions was binned into 25 ms bins yielding an ntrials x 100 bins x nunits sized matrix for each task and each day. Each unit's activity was then z-scored by subtracting by its mean and dividing by its standard deviation over trials and bins. Then, each bin and its previous two bins were concatenated to yield a ntrials x 98 chunks x (nneurons x 3) sized matrix (first two bins discarded as there were not enough lagged bins). Here, a chunk is three bins. Next, each chunk on each trial was labeled as pre-movement onset (PreMO) or post-movement onset (PostMO) if the last of the three bins in the chunk occurred prior to or following movement onset (calculated same way as in Movement onset time calculation). Each chunk on each trial was also labeled as on-beta or off-beta based on whichever label was the majority of the three bins in the chunk.

A within-day logistic regression classifier was trained from CO spike data and actual labels (PreMO, PostMO) to predict labels (using sklearn.linear_models.LogisticRegression, *Bundy et al., 2016*). To assess the probability of each data point occurring within each class the following equation was used:

$$p(y_i = 1) = \frac{1}{1 + e^{\{\beta_0 + \beta_1 * X_i\}}}, \ \ p(y_i = 0) = 1 - p(y_i = 1)$$

where $\beta_0$ and $\beta_1$ are the intercept and neural weights found by the logistic regression classifier, respectively.

The probability of a single class (PostMO) was used to assess differences in predicted probabilities for chunks. Typically in logistic regression, a threshold of 0.5 is used to classify the two classes, where observations with $p(y_i = 1)$ greater than 0.5 would be assigned the label of '1' and $p(y_i = 1)$ less than 0.5 would be assigned the label of '0'. Training with unbalanced groups can result in other threshold values being optimal, which are typically discovered with an ROC curve analysis (*Bradley, 1997*). We found that optimal thresholds for maximizing percent correct classification wre 0.5 and 0.315 for Monkey G and Monkey C. These values are the MO thresholds in *Figure 7*.

## Acknowledgements

We thank V. Athalye, S. Gowda, H. Moorman, K. So, and A. Orsborn for technical assistance.

## Additional information

### Funding

| Funder | Grant reference number | Author |
| --- | --- | --- |
| National Science Foundation | GRFP | Preeya Khanna |
| Defense Sciences Office, DARPA | W911NF-14- 2-0043 | Jose M Carmena |

The funders had no role in study design, data collection and interpretation, or the decision to submit the work for publication.

### Author contributions

PK, Conceptualization, Data curation, Software, Formal analysis, Validation, Investigation, Visualization, Methodology, Writing—original draft, Writing—review and editing; JMC, Conceptualization, Formal analysis, Supervision, Funding acquisition, Investigation, Methodology, Writing—original draft, Project administration, Writing—review and editing

### Author ORCIDs

Preeya Khanna, http://orcid.org/0000-0001-6402-6486
Jose M Carmena, http://orcid.org/0000-0002-0214-2489

### Ethics

Animal experimentation: All procedures were conducted in compliance with the NIH Guide for the Care and Use of Laboratory Animals and were approved by the University of California, Berkeley Institutional Animal Care and Use Committee (protocol AUP-2014-09-6720)

## Additional files

### Supplementary files

• Supplementary file 1. Supplemental methods.

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
