## [Decision Letter]

Thank you for submitting your article "β Band Oscillations in Motor Cortex Reflect Neural Population Signals that Delay Movement Onset" for consideration by *eLife*. Your article has been reviewed by three peer reviewers, one of whom is a member of our Board of Reviewing Editors, and the evaluation has been overseen by David Van Essen as the Senior Editor. The following individual involved in review of your submission has agreed to reveal his identity: Eric Maris (Reviewer #3).

The reviewers have discussed the reviews with one another and the Reviewing Editor has drafted this decision to help you prepare a revised submission.

Summary:

This study investigates the role of β oscillations in motor control using neurofeedback training of β power and decoding of spike rates across multiple simultaneously recorded neurons. Previous studies have shown that spontaneous fluctuations in β power correlate with several parameters of movement initiation, with higher β power generally slowing movement initiation. Previous studies have also used brain stimulation approaches to provide further evidence for this. The current manuscript adds substantial new evidence by using neurofeedback to modulate β power. Neurofeedback is an alternative way to experimentally control β power. It has the disadvantage that β power modulation is not pure, but accompanied by power changes in other frequency bands; yet it has the advantage over conventional brain stimulation techniques of avoiding potentially unphysiological conditions. Using a novel neurofeedback-reaching task, animals first adjust the LFP-β power in a neurofeedback task to four predetermined states, ranging from very low to very high, and then execute a reach task to a visual target. It was found that reaches preceded by low β power had a significantly faster reaction time and the population state of the neural network was significantly closer to a movement onset state than when the β power was high, demonstrating a correlation of neural population activity with longer reaction times during increased β oscillation activity. Employing neurofeedback to voluntarily modulate β LFP power in specific brain areas and then studying behavioral and neural population effects is highly novel and interesting. In summary, the study provides an interesting and substantial scientific advance. There are several points that need to be addressed to further improve the manuscript.

Results:

Strategies for variation of β power. The authors argue that subjects may choose their own subject-specific strategy for generating or suppressing β activity, including several possible 'internal behaviors'. However, can the authors sufficiently rule out 'external behavior'? In other words, could β power be modulated by more or less overt motor activity, like below-threshold movements or muscle co-contractions? This would be important to check, e.g., by sampling some EMG activity.

Why were the reach movements performed only in one direction? This makes the task highly monotonic and predictable. Animals probably have been trained before to perform the CO task in several directions. Why was it not possible to maintain multiple reach directions in the task? On the one hand, the statistics would probably not have been so conclusive with a smaller number of trials per reach condition. On the other hand, showing the effect of β power in behavior and neural network independent from the intended action would have been much more powerful. This should be properly discussed.

The authors use a complex experimental approach to deal with the fact that non-β frequencies are co-modulated during the β neurofeedback epoch of the task. A much simpler approach (which does not require new data) would be a multiple regression analysis in which one quantifies how much of the variance in the movement onset times can be explained by β power on top of the power in the frequencies below 10 Hz. (Frequencies higher than β may be ignored, because these will only have a small effect on the β power normalization, as a result of the 1/f scaling of the power.)

Results, paragraph two: This paragraph gives the impression that the β power changes were unrelated to power changes in other frequency bands. This is not convincing, because Figure 2 shows that monkeys C and G modulate their low-frequency power opposite to the β power. These spectra show z-scored power, in which the power in each frequency bin is normalized by subtracting the mean and dividing by the SD of that frequency bin across trials. However, the normalization for the feedback training was done differently: it was performed directly on the raw power values. Low-frequencies have the largest power and will therefore have the strongest impact on the normalization during feedback training. It is therefore a powerful strategy for the monkeys to modulate their low frequency power. This is precisely what monkeys C and G seem to have done. The authors should fully acknowledge that β power changes are accompanied by systematic changes in other frequency ranges. Even if they had trained monkeys on the non-normalized β power, the animals would have likely learned to co-modulate other frequency bands. Physiologically, fluctuations in power of different frequency bands are positively or negatively correlated, depending on the respective bands. Nevertheless, monkey S has apparently used a different strategy, because Figure 2 suggests that low-frequency power is not modulated opposite to β power. The authors could elaborate on this monkey's data and probably use it to demonstrate that normalized β power was enhanced during feedback training in different ways, but always had the same behavioral effect.

Subsection “Using other methods to compute β power shows the same movement onset relationship”: This is an extra test, which renders the correlation between γ power and MOT insignificant. This same test is not done for any of the other frequency bands, so we do not know whether it would affect them in the same or a different way. Also, here the individual significance testing per animal is a problem. All three subject show negative z-values, two show individually significant effects, only one misses the significance. Is the effect significant, when the data are combined across subjects?

Power in the 65-100 Hz band is typically very small in absolute terms, when compared to β power or even lower-frequency power. Thus, it likely did not contribute very much to the monkeys' strategies during feedback training. I suggest the authors quantify the percentage of 65-100 Hz power contribution to the denominator used for normalization during feedback training. I expect that this number is very small, and that the animals modulate γ power opposite to β power, simply because this is the physiological pattern. This should be acknowledged rather than argued away. It would be interesting to see whether β power and 65-100 Hz power is similarly anticorrelated outside the feedback task, e.g. during CO task performance.

There is the need for further clarification regarding the logistic regression classifier to demonstrate (1) the relation between the population firing rates and movement and (2) how this relation is modulated by β oscillatory power. It is possible that this relation is driven by a small fraction of the recorded neurons, and it would be useful to know this. There are regression techniques (forward, backward, stepwise predictor selection) that provide this information. Given the identification of prediction-relevant neurons, it would be interesting to know exactly how β oscillatory power modulates their activity. Provided the number of prediction-relevant neurons is not too large, this should not be too difficult.

Was there a significant correlation between time to target and movement onset time, when data from all animals are combined?

Where the authors compare neuronal activity during CO and NF tasks, they should acknowledge that the CO task occurs always after the NF task, such that there could be an unspecific effect of time.

Subsection “Modified β neurofeedback task shows 1-10 Hz band power does not account for movement onset time trend”: "…movement onset times followed […] not 1-10 Hz power ordering." For the xy control task, please provide a test for the low frequency band stating the absence of this effect.

Subsection “Comparing single and multi unit responses to β oscillations during the CO and NR tasks” paragraph four: Are those smaller R values significant? Is the difference between within-task slopes and across-task slopes significant?

Paragraph six of the same subsection: "…R2 values less than 0.5 in Figure 6." Are these values still significant, i.e., when tested by a Monte Carlo test?

Figure 4 shows a cloud of points (small x-axis values and large y-axis values) that stand apart from the main cloud. Are they included in the analysis? Please comment.

Figure 7: The values for monkey G are about an order of magnitude larger than the values for monkey C. The effects look very similar. The authors should nevertheless comment on the huge difference in magnitude.

Figure 7: Is this significant after combining monkeys?

Methods:

Calculating the LFP power with the multi-taper method with K=5 tapers from a time window of T=0.2s (see Materials and methods) leads to spectral smoothing with a half-bandwidth W=15 Hz!!! This follows from the equation K=2TW-1 (see your multi-taper method tutorial for reference). This means that your power estimate at frequency f is actually represents the smoothed power from the frequency band [f-15, f+15] Hz. However, this does not seem to be the case, when looking a Figure 1. Using fewer tapers would reduce frequency smoothing at the expense of more noise. However, since "β_est" was averaged in the frequency range of 25-40 Hz, more noise might not be so problematic. Please clarify what was actually done.

Results are only based on p-values and no effect-size measures are reported. These can easily be provided and are useful (see van Ede et al., 2012, JNPhys, for a related study and a motivation). These effect-size measures could be the percentage of explained variance for the movement onset times (for the first observation) and the probability of correctly classifying a spiking pattern as coming from the pre or the post movement period (for the second observation).

Authors mainly report statistical tests separately for the three monkeys. This should always be accompanied by a test that combines across monkeys. This is particularly relevant, as they also report that some effects are not consistent across subjects. In that case, the decisive test is the one that combines data across subjects.

On a number of occasions, the authors perform a statistical test on the outcome of a classification in which training and test sample were identical. Such a test is biased. Although biased, this does not invalidate the paper's main results, which were obtained with a different training and test sample. This issue should be explicitly addressed.

Results section, paragraph three: "…3-10 days of executing the NR task." This suggests, there are only 3-10 sessions of data per animal, but Figure 1 indicates a much larger number of sessions per animal. Please clarify! Were there multiple sessions per day?

Subsection “NR task controls”: Trials with MOT smaller than 0 and greater than 0.7 s were removed. Was the same pruning done for the target analysis? This should be consistent across analyses.

Subsection “Movement onset trend is specific to β band frequencies”: The analysis should not only be performed on pre-defined bands, but correlations between power and MOT should be calculated for each frequency bin. This is not a must, just a suggestion. In the current frequency-bin definition, the 1-10 Hz bin combines δ, theta and α. The authors should at least make an effort to separate theta and α.

The authors should explain how they derive channel level activity, rather than merely pointing to the paper of Chestek et al., 2011.

Subsection “Comparing single and multi unit responses to β oscillations during the CO and NR task”: This will likely distort results.

In the same section: Is 1.5 sec before go cue not too much for the CO task? Please explain!

Materials and methods section: Please also specify the length of the electrodes.

Also please be more specific on the implantation location of the electrodes, e.g., hand or reach area of M1 and PMd?

Subsection “Chance level performance of β neurofeedback epoch”: Besides the z-score, please also provide some absolute numbers on the performance, e.g., like: "on average, xx successful trials per minute with an overall success rate of yy% ".

Subsection “NR task variant using β band and 1-10 Hz”: Same dpss as used before?

Subsection “Trial re-allocation analysis”: Authors might want to consider this paper, which reveals issues arising when data are binned before calculating correlations: A jackknife approach to quantifying single-trial correlation between covariance-based metrics undefined on a single-trial basis. Richter CG, Thompson WH, Bosman CA, Fries P. Neuroimage. 2015 Jul 1;114:57-70. doi: 10.1016/j.neuroimage.2015.04.040. This does not render r-values obtained through binning invalid, however, their absolute values are typically inflated.

Figure 6: Correlation coefficients might be more appropriate than a linear regression (R2), since the values on the x- and y-axis are 'equally' independent variables.

Figure 6: If those slopes are to be compared to the subset based slopes, they also need to be based on subsets.

Writing and presentation:

It would be helpful if the first paragraph of the Results section would specify the recorded brain areas, even if this is mentioned elsewhere.

Throughout, the authors use the word "trend" to mean a significant systematic dependence. I guess this is motivated by their use of Cuzick's Test for trend. Readers might misunderstand this, because the word "trend" is often used for an effect that does not reach significance.

Introduction section, final paragraph: It should be mentioned here that monkeys might modulate normalized β power by modulating power outside the β band, i.e. by modulating the denominator. Also mention here that this will be addressed later in more detail.

Subsection “The neurofeedback epoch results in varying levels of initial β power prior to reaching”: The authors should emphasize more that they relate behavioral performance to β targets, not the actual β levels. That is a crucial difference to previous studies, which related behavioral performance to spontaneously occurring β levels. The β levels observed in the present study were likely highly correlated with β targets. However, it is conceptually important that the authors intervene with the system through neurofeedback by setting β targets. This is a crucial distinction to previous related studies and should be made more explicit.

Subsection “Modified β neurofeedback task shows 1-10 Hz band power does not account for movement onset time trend” "within the bin": Do the authors actually mean speed within each bin, or rather the speed during the response?

In the same subsection: Neither "four bins in a row" is clear, nor does the reader now the particular units that are referred to.

Subsection “Comparing single and multi unit responses to β oscillations during the CO and NR 338 tasks”: What is the alignment in Figure 6, upper and lower panels?

Subsection “Β oscillations during the NR task and CO task reflect a shift in neural population activity away from a movement onset state”: The authors need to mention that they compare pre-MO to post-MO in the NR task. This is clear from the figure, but should also be clear in the main text.

In the same subsection: The lines leading up to the description of Figure 7. This figure shows distances of the classifier to MO threshold as a function of β target, not as a function of observed β level. The feedback-related definition of β targets is the crucial manipulation in this study. Therefore, the authors should here point to the potential influence of β targets (not β levels) on classifier distances.

In the same subsection, paragraph two: Consider replacing 'lag' with 'history'.

Also in this subsection: "… how far an observation's neural state was [away] from [the state of] movement onset." Consider including the [bracketed] words for improved clarity.

In paragraph five of this section, and also multiple times later: 'NF trials': Do you mean NR trials? Please be consistent throughout the MS and the figures!

Subsection “Advantages of using neurofeedback to study behavioral correlates”: "the neural signal is not tarnished with a stimulation artifact". This comes quite suddenly. The authors should introduce that β could be modulated through tACS or similar methods, and that this poses challenges for simultaneous recording of neural activity.

Subsection “Calculation of β neurofeedback cursor”: This description with new neural data and previous data was confusing for me.

Subsection “Chance level performance of β neurofeedback epoch”: "chance fluctuations": Specify that this is about chance fluctuations in β power.

Subsection “Trial re-allocation analysis”: "assess whether non-normalized β power" should better read "assess whether MOTs for non-normalized β power".

Subsection “Β amplitude to spike rate mapping”: The authors use the word "bin" in two meanings, one of them referring to something like "compound bins" made of three smaller bins. This is confusing and should be reworded.

Subsection “Classification between neurofeedback and reaching tasks”: Please provide a reference here to explain the used method.

Figure 1: The colors for the different monkeys are hard to separate. If readers print this figure in greyscale, they will be hardly distinguishable. It would also be helpful if different symbols were used (e.g. open and closed circles) for training versus task/recording sessions, respectively.

Figure 1: Some x-axis ticks would be useful here.

Figure 2: Is this data from one session per animal? Please specify!

Figure 2 would benefit greatly from a color legend inside the figure (rather than merely in the legend). Also, the authors use "teal", which some non-native English speakers do not know. They could circumvent this potential source of confusion by using "green".

Figure 3: What is the sample size here?

Figure 5: The title "Non-β frequencies do not account for the movement onset trend" does not follow from the data presented here (1-10 Hz could also account for it), but only from later control analysis. Therefore, please change the figure title.

Figure 6: The figure would benefit from a color legend inside the figure. Generally, more (color) labeling inside the figures would make it easier to understand the figures.

Figure 7: In subpanel labels, 'NF' should probably read 'NR'.

Legend. "Same colormap as in Figure 6", (not F-H).

Figure 7: Blue and green traces are not ordered as predicted. This should be mentioned.

---

## [Author Response]

*[…] Results:*

*Strategies for variation of β power. The authors argue that subjects may choose their own subject-specific strategy for generating or suppressing β activity, including several possible 'internal behaviors'. However, can the authors sufficiently rule out 'external behavior'? In other words, could β power be modulated by more or less overt motor activity, like below-threshold movements or muscle co-contractions? This would be important to check, e.g., by sampling some EMG activity.*

Thank you for the comment. First, we did video two NHPs (Monkeys C, G) completing the NR task. In both videos the left hand is visible, and in the second video the left hand, full trunk and both legs of the animal are visible. During the neurofeedback epochs of the task, neither animal is making perceptible movements. Second, a previous study in NHPs investigating neurofeedback of the motor cortical LFP in the 30-43 Hz frequency band, a nearly overlapping frequency range to this study, showed no movements nor EMG activity while monkeys performed neurofeedback (Engelhard et al., 2013). This evidence makes us more confident that monkeys likely did not engage an overt motor strategy while performing neurofeedback.

Finally, even if the monkeys did make overt movements to assist in performance of the neurofeedback epoch, the movements would be most likely occur during the lowest β target since movement desynchronizes β power. However, previous studies have shown that contralateral limb movements (in this case, left limb movements) lengthen a simple reaction time performed (Begeman et al., 2007; Buenaventura and Sarkin, 1996). This evidence suggests that if the animals are performing some movements to make the β cursor lower, their subsequent right arm reaches likely are slower than they would be without the extraneous movements. Another possibility is that subjects are using isometric contraction to increase β power, as demonstrated in (Baker et al., 1997). However, one study shows that if the hand of interest (here, right hand) is isometrically contracted, there is no significant difference in movement onset time for this condition compared to movement onset for a control condition when the subject is at rest (Castellote et al., 2004). This evidence suggests that if the subject is moving to lower the β cursor, these movements would lengthen their arm-reaching movement onset time, likely making the effect we observe *less* pronounced than it would be without the extra movements.

This point has been appended to the discussion (‘alternative explanations’).

*Why were the reach movements performed only in one direction? This makes the task highly monotonic and predictable. Animals probably have been trained before to perform the CO task in several directions. Why was it not possible to maintain multiple reach directions in the task? On the one hand, the statistics would probably not have been so conclusive with a smaller number of trials per reach condition. On the other hand, showing the effect of β power in behavior and neural network independent from the intended action would have been much more powerful. This should be properly discussed.*

Thank you for this comment. We did try to train one monkey to perform the NR task where the reach target was not predictable. When doing the NR task with unprepared reaches, there were no significant differences in MOT by β target. Likely, this effect was due to the fact that unprepared reaches have longer MOTs than prepared reaches (sometimes as big a difference as 60 ms found in Monkey G in Churchland et al., 2006) and that differential levels of β power quickly go away after the subjects have held their β cursor in the β target (as seen in Figure 2 following the first vertical red line). Thus, the β levels of the NHPs could have been equivalent when they initiated their unprepared reaches since the time to prepare and initiate movement took longer in the case of unprepared reaches. Another challenge with instructing unprepared reaches is that cue presentation has been associated with increases in β power (Leventhal et al., 2012; Saleh et al., 2010). Thus, if the reach is unprepared, providing a cue to the animal after the neurofeedback epoch has been completed may equalize the subjects’ β state. We do expect that our findings would generalize to unprepared movements, though the current paradigm may not be able to test that hypothesis directly.

This point has been appended to the discussion (‘alternative explanations’).

*The authors use a complex experimental approach to deal with the fact that non-β frequencies are co-modulated during the β neurofeedback epoch of the task. A much simpler approach (which does not require new data) would be a multiple regression analysis in which one quantifies how much of the variance in the movement onset times can be explained by β power on top of the power in the frequencies below 10 Hz. (Frequencies higher than β may be ignored, because these will only have a small effect on the β power normalization, as a result of the 1/f scaling of the power.)*

Thank you for this suggestion. We performed an analysis of R^2^ when fitting linear models predicting MOT with β power, low frequency power, or both variables. We find that β frequency (25-40 Hz) power accounts for more MOT variance than low frequency (1-10 Hz) power. When comparing two models that predict MOT (MOT ~ Low Freq versus MOT ~ Low Freq + Β), the F-test demonstrates adding β power significantly improves model performance for all animals. See Table 3.

*Results, paragraph two: This paragraph gives the impression that the β power changes were unrelated to power changes in other frequency bands. This is not convincing, because Figure 2 shows that monkeys C and G modulate their low-frequency power opposite to the β power. These spectra show z-scored power, in which the power in each frequency bin is normalized by subtracting the mean and dividing by the SD of that frequency bin across trials. However, the normalization for the feedback training was done differently: it was performed directly on the raw power values. Low-frequencies have the largest power and will therefore have the strongest impact on the normalization during feedback training. It is therefore a powerful strategy for the monkeys to modulate their low frequency power. This is precisely what monkeys C and G seem to have done. The authors should fully acknowledge that β power changes are accompanied by systematic changes in other frequency ranges.*

Thank you, please see section ‘The neurofeedback epoch results in varying levels of initial β power prior to reaching’ were we added the following clarification:

“Since calculation of the β cursor position involved an estimate of broadband power, changes in non-β frequencies also affected β cursor position. In some subjects (Monkeys C, G), increases and decreases in β power were accompanied with reliable decreases and increases in low frequencies (1-10 Hz).”

Also note the section titled “Non-β frequencies are co-modulated during the β neurofeedback epoch of the task” (first line):

“Since the neurofeedback epoch required control of normalized β power, it is possible for subjects to have neurofeedback strategies that involve modulation of non-β frequency bands, in addition to the β band, to move the cursor.”

This also explains that non-β band power changes can influence cursor position.

*Even if they had trained monkeys on the non-normalized β power, the animals would have likely learned to co-modulate other frequency bands. Physiologically, fluctuations in power of different frequency bands are positively or negatively correlated, depending on the respective bands. Nevertheless, monkey S has apparently used a different strategy, because Figure 2 suggests that low-frequency power is not modulated opposite to β power. The authors could elaborate on this monkey's data and probably use it to demonstrate that normalized β power was enhanced during feedback training in different ways, but always had the same behavioral effect.*

We agree that Monkeys S’s low frequencies are not modulated as drastically as Monkey C and Monkey G’s, and point out this difference in the Discussion section: Advantages of using neurofeedback to study behavioral correlates:

“For example, while Monkeys C and G inversely modulated low frequency (1-10 Hz) power with β frequency power, Monkey S did not modulate 1-10 Hz power as drastically but did exhibit increased 50-60 Hz power with increased β power (Figure 2).”

*Subsection “Using other methods to compute β power shows the same movement onset relationship”: This is an extra test, which renders the correlation between γ power and MOT insignificant. This same test is not done for any of the other frequency bands, so we do not know whether it would affect them in the same or a different way. Also, here the individual significance testing per animal is a problem. All three subject show negative z-values, two show individually significant effects, only one misses the significance. Is the effect significant, when the data are combined across subjects?*

We decided not to include the results of this test. Instead, we choose to perform a reviewer suggested analysis to show that the percentage of movement onset time variance accounted for by the normalized γ power is much lower than is accounted for by β power. Further, when comparing a model that predicts MOT from γ power versus a model that predicts MOT from γ and β power, for all animals we find that the second model is significantly better, suggesting that knowledge of β power adds predictive power that is not captured in knowledge of γ power. See Table 3.

*Power in the 65-100 Hz band is typically very small in absolute terms, when compared to β power or even lower-frequency power. Thus, it likely did not contribute very much to the monkeys' strategies during feedback training. I suggest the authors quantify the percentage of 65-100 Hz power contribution to the denominator used for normalization during feedback training. I expect that this number is very small, and that the animals modulate γ power opposite to β power, simply because this is the physiological pattern. This should be acknowledged rather than argued away. It would be interesting to see whether β power and 65-100 Hz power is similarly anticorrelated outside the feedback task, e.g. during CO task performance.*

Indeed, we find that the contribution of γ power to the denominator of the cursor computation (total broadband power) is very small (see Table 4).

We also add the following to the result section titled“Non-β frequencies are co-modulated during the β neurofeedback epoch of task”:

“Indeed, β power and γ power have been shown to be anti-correlated in motor-related regions during tasks involving movement (Courtemanche et al., 2003; Schoffelen et al., 2005), in prefrontal cortex during working memory tasks (Lundqvist et al., 2016), and in parkinsonian subjects at rest (Fogelson et al., 2005). Increased γ power may then be a physiological pattern that emerges with reduced β power. It is unlikely that subjects are relying on changes in γ power, which would change the denominator term in the β cursor computation, to drive their neurofeedback strategy since γ power constitutes less than 3% of the total broadband estimate (Table 4).”

*There is the need for further clarification regarding the logistic regression classifier to demonstrate (1) the relation between the population firing rates and movement and (2) how this relation is modulated by β oscillatory power. It is possible that this relation is driven by a small fraction of the recorded neurons, and it would be useful to know this. There are regression techniques (forward, backward, stepwise predictor selection) that provide this information. Given the identification of prediction-relevant neurons, it would be interesting to know exactly how β oscillatory power modulates their activity. Provided the number of prediction-relevant neurons is not too large, this should not be too difficult.*

In addition to performing a population decoding analysis as in Figure 7, we also analyzed individual units that exhibited 1) the ability to discriminate PreMO and PostMO accurately on held-out test data in both tasks, and 2) showed significantly greater selection of PreMO during β-events. Units showing both properties were termed ‘chosen units’ and other were termed ‘unchosen units’.

1) ‘Chosen Units’ compared to ‘unchosen units’ were found to:

a) Increase in firing rate with movement onset

b) Decrease in firing rate with increased β amplitude

c) Have higher mean firing rates overall

d) Have no significant difference in modulation compared to unchosen units

e) No significant difference in β rhythmicity during spiking

While most ‘chosen units’ exhibited increase in firing with movement compared to hold periods, some did show reduced firing rate during movement. We compared ‘positive chosen units’ (units with increased firing during movement) to ‘negative chosen units’ (chosen units with decreased firing rate during movement. We find:

a) Increased firing with increased β amplitude in the ‘negative chosen units’

b) No significant difference in modulation during task

c) Not significant but slight increase in β rhythmicity in the 20 – 45 Hz range (5 Hz extension of the 25 – 40 Hz range used in the task) in ‘negative chosen units’.

These finding are summarized in Extended Figure 7, Extended Data Table 1, and in the Results section.

*Was there a significant correlation between time to target and movement onset time, when data from all animals are combined?*

Yes there is. However, if we subtract out the predicted MOTs based on time to target and do a 2-tailed Cuzick’s test on the residual MOTs, we still see a significantly increasing MOT with increased β target. This has been added to the manuscript in Results section “Β target acquisition difficulty does not correlate with movement onset time”.

*Where the authors compare neuronal activity during CO and NF tasks, they should acknowledge that the CO task occurs always after the NF task, such that there could be an unspecific effect of time.*

This is stated in section “comparing single and multi unit responses to β oscillations during the CO and NR tasks”:

“On most days, subjects performed 5-10 minutes of the CO task prior to beginning the NR task.”

*Subsection “Modified β neurofeedback task shows 1-10 Hz band power does not account for movement onset time trend”: "…movement onset times followed […] not 1-10 Hz power ordering." For the xy control task, please provide a test for the low frequency band stating the absence of this effect.*

A one-tailed Cuzick’s test for ordering of β targets by 1, 2, 3, 4 was added. If high δ power was driving fast RTs, then we expect the order of RTs to best be described by 1, 2, 3, 4 since this is the descending order of power in 1-10Hz. When the one-way Cuzick’s test for increasing RT with decreasing δ power is performed though, we find an insignificant trend (z = -2.5713, p = 0.99493). This has now been added to the Results section “*Modified β neurofeedback task shows 1-10 Hz band power does not account for movement onset time increase”.*

*Subsection “Comparing single and multi unit responses to β oscillations during the CO and NR tasks” paragraph four: Are those smaller R values significant? Is the difference between within-task slopes and across-task slopes significant?*

This has now been addressed by completing a paired Student’s t-test assessing differences in slopes between CO_1_ – CO_2,_ NF_1_ -- NF_2_, and CO_2_ – NF_1_. Units were combined across days. See section “Comparing single and multi unit responses to β oscillations during the CO and NR tasks”.

*Paragraph six of the same subsection: "…R2 values less than 0.5 in Figure 6." Are these values still significant, i.e., when tested by a Monte Carlo test?*

This line has been removed.

*Figure 4 shows a cloud of points (small x-axis values and large y-axis values) that stand apart from the main cloud. Are they included in the analysis? Please comment.*

No, all trials with MOT > 0.7 sec are removed. The figure has been updated to remove these points.

*Figure 7: The values for monkey G are about an order of magnitude larger than the values for monkey C. The effects look very similar. The authors should nevertheless comment on the huge difference in magnitude.*

The following line has been added to Results section: “Β oscillations during the NR task and CO task reflect a shift in neural population activity away from a movement onset state”:

“Note that Monkey G does exhibit about an order of magnitude more reliable separation between PreMO and PostMO than Monkey C (y axis in Figure 7), and this is likely due to the lower neural signal quality in Monkey C (implanted ~3 years prior to study without resolvable single or multi-units) than Monkey G (implanted only ~1 year prior to study, with resolvable single and multi-units).”

*Figure 7: Is this significant after combining monkeys?*

Yes – all tests now combine across monkeys.

*Methods:*

*Calculating the LFP power with the multi-taper method with K=5 tapers from a time window of T=0.2s (see Materials and methods) leads to spectral smoothing with a half-bandwidth W=15 Hz!!! This follows from the equation K=2TW-1 (see your multi-taper method tutorial for reference). This means that your power estimate at frequency f is actually represents the smoothed power from the frequency band [f-15, f+15] Hz. However, this does not seem to be the case, when looking a Figure 1. Using fewer tapers would reduce frequency smoothing at the expense of more noise. However, since "β_est" was averaged in the frequency range of 25-40 Hz, more noise might not be so problematic. Please clarify what was actually done.*

The methods are correct, and 5 tapers were indeed used with a time-bandwidth product of 3. As you point out, this results in substantial smoothing during the online estimate of β power. Figure 1 was computed using the multi-taper method, but was also ‘detrended’ to remove the 1/f trend in typical neurological PSDs. This is now noted in the figure legend and Materials and methods section ‘Power spectral densities (PSDs)’. Attached here is a comparison of the non-detrended spectrogram and the detrended spectrogram seen in Figure 1.

*Results are only based on p-values and no effect-size measures are reported. These can easily be provided and are useful (see van Ede et al., 2012, JNPhys, for a related study and a motivation). These effect-size measures could be the percentage of explained variance for the movement onset times (for the first observation) and the probability of correctly classifying a spiking pattern as coming from the pre or the post movement period (for the second observation).*

See Table 3 for amount of variance accounted for. We also report percent of correctly classifying held-out CO and NF spike patterns in the manuscript in Results section “Β oscillations during the NR task and CO task reflect a shift in neural population activity away from a movement onset state”.

*Authors mainly report statistical tests separately for the three monkeys. This should always be accompanied by a test that combines across monkeys. This is particularly relevant, as they also report that some effects are not consistent across subjects. In that case, the decisive test is the one that combines data across subjects.*

Now all reported statistical tests include tests combined across monkeys.

*On a number of occasions, the authors perform a statistical test on the outcome of a classification in which training and test sample were identical. Such a test is biased. Although biased, this does not invalidate the paper's main results, which were obtained with a different training and test sample. This issue should be explicitly addressed.*

This is has been corrected. In Figure 7 only held-out test data is reported. In Figure 7 both training data and test data is used to compare the classification scores of spiking patterns from on-β and off-β events. This is done because the classifier has no knowledge of which spiking patterns are on-β and off-β, so it is not explicitly trained.

*Results section, paragraph three: "…3-10 days of executing the NR task." This suggests, there are only 3-10 sessions of data per animal, but Figure 1 indicates a much larger number of sessions per animal. Please clarify! Were there multiple sessions per day?*

Yes, there were multiple 10-40 minute sessions per day. This has now been clarified in the manuscript.

*Subsection “NR task controls”: Trials with MOT smaller than 0 and greater than 0.7 s were removed. Was the same pruning done for the target analysis? This should be consistent across analyses.*

Yes, this was also done for all analyses except for those in Figure 2.

*Subsection “Movement onset trend is specific to β band frequencies”: The analysis should not only be performed on pre-defined bands, but correlations between power and MOT should be calculated for each frequency bin. This is not a must, just a suggestion. In the current frequency-bin definition, the 1-10 Hz bin combines δ, theta and α. The authors should at least make an effort to separate theta and α.*

Done. Please see end of section “Modified β neurofeedback task shows 1-10 Hz band power does not account for movement onset time increase”.

*The authors should explain how they derive channel level activity, rather than merely pointing to the paper of Chestek et al., 2011.*

This has been added to Materials and methods section.

*Subsection “Comparing single and multi unit responses to β oscillations during the CO and NR task”: This will likely distort results.*

We are not sure to what the reviewer is referring too. Some of the line number references have been off compared to the uploaded version on *eLife* portal. In our version of the submitted manuscript this line reads “We assess whether units fire at similar rates during CO task β oscillations and NR…”.

*In the same section: Is 1.5 sec before go cue not too much for the CO task? Please explain!*

Only bins where the hand was still were used for analysis in Figure 6 and Figure 7. You’re correct in noting that 1.5 seconds prior to the go cue is too long and would capture more time than just the hold period, but only the non-movement part of the 1.5 seconds is used, which just captures the hold period. Prior to the hold period the hand was being brought to the center target so these time points would not satisfy the ‘non-movement’ requirement to be included in the analysis. This method was used to account for the variable hold times in the CO task. Note that Figure 6 used the full 2.5 second trial, which included activity before the hold period, the hold period, the reach, and any activity after the reach.

*Materials and methods section: Please also specify the length of the electrodes.*

6.5 mm. This has been added to the Materials and methods section.

*Also please be more specific on the implantation location of the electrodes, e.g., hand or reach area of M1 and PMd?*

Arm area of M1/PMd – has been added to the Materials and methods.

*Subsection “Chance level performance of β neurofeedback epoch”: Besides the z-score, please also provide some absolute numbers on the performance, e.g., like: "on average, xx successful trials per minute with an overall success rate of yy% ".*

Done, added to beginning of results.

*Subsection “NR task variant using β band and 1-10 Hz”: Same dpss as used before?*

Yes.

*Subsection “Trial re-allocation analysis”: Authors might want to consider this paper, which reveals issues arising when data are binned before calculating correlations: A jackknife approach to quantifying single-trial correlation between covariance-based metrics undefined on a single-trial basis. Richter CG, Thompson WH, Bosman CA, Fries P. Neuroimage. 2015 Jul 1;114:57-70. doi: 10.1016/j.neuroimage.2015.04.040. This does not render r-values obtained through binning invalid, however, their absolute values are typically inflated.*

Thank you for the reference. We solely report r-values in Figure 6 as a summary of the plot and do not do statistics on the r-values. Instead, we directly do a t-test on slope calculations which use very small bins (1ms). In Figure 6, we do bin spikes and compare the distribution of average firing rate across tasks for each unit using the Mann-Whitney test, but do not do regressions.

*Figure 6: Correlation coefficients might be more appropriate than a linear regression (R2), since the values on the x- and y-axis are 'equally' independent variables.*

Thanks for the suggestion. We now list correlation coefficients instead of R^2^.

*Figure 6: If those slopes are to be compared to the subset based slopes, they also need to be based on subsets.*

Agreed. This has been fixed. Within task assessments were done based on subsets, and across task assessments used CO_2_ – NF_1_, a subset.

*Writing and presentation:*

*It would be helpful if the first paragraph of the Results section would specify the recorded brain areas, even if this is mentioned elsewhere.*

Added.

*Throughout, the authors use the word "trend" to mean a significant systematic dependence. I guess this is motivated by their use of Cuzick's Test for trend. Readers might misunderstand this, because the word "trend" is often used for an effect that does not reach significance.*

Thank you. The word ‘trend’ has been replaced in all locations to avoid confusion.

*Introduction section, final paragraph: It should be mentioned here that monkeys might modulate normalized β power by modulating power outside the β band, i.e. by modulating the denominator. Also mention here that this will be addressed later in more detail.*

We added the following line to the second paragraph of the Results section: “Indeed, modulation of non-β frequency bands can influence the position of the β cursor through the normalization factor, a point discussed in detail further along in the study.”

*Subsection “The neurofeedback epoch results in varying levels of initial β power prior to reaching”: The authors should emphasize more that they relate behavioral performance to β targets, not the actual β levels. That is a crucial difference to previous studies, which related behavioral performance to spontaneously occurring β levels. The β levels observed in the present study were likely highly correlated with β targets. However, it is conceptually important that the authors intervene with the system through neurofeedback by setting β targets. This is a crucial distinction to previous related studies and should be made more explicit.*

The line “Other groups have found correlations between increased β power and reduced onset speed, peak speed, and onset acceleration (see Joundi et al., 2012; Pogosyan et al., 2009) which we do not find consistently across subjects (see Table 2)” has been changed to “Other groups have found correlations between increased β power and reduced onset speed, peak speed, and onset acceleration (see Joundi et al., 2012; Pogosyan et al., 2009) which we do not find consistently across subjects when comparing metrics grouped based on preceding β target (see Table 2). “

Subsection “Modified β neurofeedback task shows 1-10 Hz band power does not account for movement onset time trend” "within the bin": Do the authors actually mean speed within each bin, or rather the speed during the response?

We actually mean speed within each bin. We want to select bins that are ‘slow’ (prior to movement) and ‘fast’ (during movement).

*In the same subsection: Neither "four bins in a row" is clear, nor does the reader now the particular units that are referred to.*

This sentence has been fixed for clarity.

*Subsection “Comparing single and multi unit responses to β oscillations during the CO and NR 338 tasks”: What is the alignment in Figure 6, upper and lower panels?*

Aligned to the start of a ‘β-on’ event. This has been clarified in the figure legend and text.

*Subsection “Β oscillations during the NR task and CO task reflect a shift in neural population activity away from a movement onset state”: The authors need to mention that they compare pre-MO to post-MO in the NR task. This is clear from the figure, but should also be clear in the main text.*

This has been clarified.

*In the same subsection: The lines leading up to the description of Figure 7. This figure shows distances of the classifier to MO threshold as a function of β target, not as a function of observed β level. The feedback-related definition of β targets is the crucial manipulation in this study. Therefore, the authors should here point to the potential influence of β targets (not β levels) on classifier distances.*

Thanks, We have clarified this point after describing Figure 7.

*In the same subsection, paragraph two: Consider replacing 'lag' with 'history'.*

Done.

*Also in this subsection: "… how far an observation's neural state was [away] from [the state of] movement onset." Consider including the [bracketed] words for improved clarity.*

Done.

In paragraph five of this section, and also multiple times later: 'NF trials': Do you mean NR trials? Please be consistent throughout the MS and the figures!

Done in the manuscript and Figure 7.

*Subsection “Advantages of using neurofeedback to study behavioral correlates”: "the neural signal is not tarnished with a stimulation artifact". This comes quite suddenly. The authors should introduce that β could be modulated through tACS or similar methods, and that this poses challenges for simultaneous recording of neural activity.*

Done, thanks.

*Subsection “Calculation of β neurofeedback cursor”: This description with new neural data and previous data was confusing for me.*

Simplified.

*Subsection “Chance level performance of β neurofeedback epoch”: "chance fluctuations": Specify that this is about chance fluctuations in β power.*

Done, thanks.

*Subsection “Trial re-allocation analysis”: "assess whether non-normalized β power" should better read "assess whether MOTs for non-normalized β power".*

Done, thanks.

*Subsection “Β amplitude to spike rate mapping”: The authors use the word "bin" in two meanings, one of them referring to something like "compound bins" made of three smaller bins. This is confusing and should be reworded.*

Done, thanks. The 3-bin is now called a ‘chunk’.

*Subsection “Classification between neurofeedback and reaching tasks”: Please provide a reference here to explain the used method.*

Done, thanks – added (Bundy et al., 2016).

*Figure 1: The colors for the different monkeys are hard to separate. If readers print this figure in greyscale, they will be hardly distinguishable. It would also be helpful if different symbols were used (e.g. open and closed circles) for training versus task/recording sessions, respectively.*

Done. No training is included in this figure.

*Figure 1: Some x-axis ticks would be useful here.*

Done

*Figure 2: Is this data from one session per animal? Please specify!*

No, averaged over all sessions. This is now clarified.

*Figure 2 would benefit greatly from a color legend inside the figure (rather than merely in the legend). Also, the authors use "teal", which some non-native English speakers do not know. They could circumvent this potential source of confusion by using "green".*

Done.

*Figure 3: What is the sample size here?*

Added to figure.

*Figure 5: The title "Non-β frequencies do not account for the movement onset trend" does not follow from the data presented here (1-10 Hz could also account for it), but only from later control analysis. Therefore, please change the figure title.*

Done.

*Figure 6: The figure would benefit from a color legend inside the figure. Generally, more (color) labeling inside the figures would make it easier to understand the figures.*

Done.

Figure 7: In subpanel labels, 'NF' should probably read 'NR'.

*Legend. "Same colormap as in Figure 6", (not F-H).*

Done.

*Figure 7: Blue and green traces are not ordered as predicted. This should be mentioned.*

Added the following line to Results section: “Β oscillations during the NR task and CO task reflect a shift in neural population activity away from a movement onset state”:

“In Figure 7, the green line corresponding to the lowest β target is further from the blue line corresponding to the mid-low β target, showing that the presence of more β oscillations in the mid-low β target is not the only factor in determining distance from the MO threshold.”